# Unified all-atom molecule generation with neural fields

**Matthieu Kirchmeyer**[1,†]   **Pedro O. Pinheiro**[1,†]   **Emma Willett**[1]   **Karolis Martinkus**[1,‡]
**Joseph Kleinhenz**[1]   **Emily K. Makowski**[2]   **Andrew M. Watkins**[1]
**Vladimir Gligorijevic**[1]   **Richard Bonneau**[1]   **Saeed Saremi**[1]

[1]Prescient Design, Genentech    [2]Antibody Engineering, Genentech

## Abstract

Generative models for structure-based drug design are often limited to a specific modality, restricting their broader applicability. To address this challenge, we introduce FuncBind, a framework based on computer vision to generate target-conditioned, all-atom molecules across atomic systems. FuncBind uses neural fields to represent molecules as continuous atomic densities and employs score-based generative models with modern architectures adapted from the computer vision literature. This modality-agnostic representation allows a single unified model to be trained on diverse atomic systems, from small to large molecules, and handle variable atom/residue counts, including non-canonical amino acids. FuncBind achieves competitive *in silico* performance in generating small molecules, macrocyclic peptides, and antibody complementarity-determining region loops, conditioned on target structures. FuncBind also generated *in vitro* novel antibody binders via *de novo* redesign of the complementarity-determining region H3 loop of two chosen co-crystal structures. As a final contribution, we introduce a new dataset and benchmark for structure-conditioned macrocyclic peptide generation[*].

## 1   Introduction

A central challenge in drug discovery is designing molecules that bind specifically to a target protein [1]. This task involves navigating a diverse landscape of molecular modalities, from small organic compounds to large biomolecules, each with unique chemical properties. Structure-based approaches are frequently employed to meet this challenge, utilizing the 3D structure of a target site (often an accessible protein region) to generate novel molecules with high affinity. Generative models are emerging as a powerful data-driven alternative to established traditional techniques such as virtual screening and physics-based simulations. These newer models can explore vast chemical spaces more effectively to identify molecules with desired binding properties [2].

Most structure-based generative models specialize on a single molecule modality to better account for specific physicochemical properties. Focusing on a single molecular modality also simplifies data gathering and augmentation, training, representation choice, and metrics used for validation. Generative models of small molecules typically represent molecules as point cloud of atoms [3, 4, 5] or discretized atomic densities [6, 7]. Most protein generative models leverage the fact that proteins are sequences of amino acids to represent them with point clouds of residues, where each residue contains multiple atoms [8], recovering their sequences with, *e.g.*, co-generation [9] or inverse folding [10]. Many protein-centric models also rely on large sequence databases and self-supervised generative models for sequence that can help in scoring and generating/proposing mutations.

---

[*]The code is available at `https://github.com/prescient-design/funcbind`. The checkpoints at `https://huggingface.co/mkirchmeyer/funcbind/`.
[†]Equal contribution, ‡work done at Genentech. Correspondence to `matthieu.kirchmeyer@gmail.com`, `pedro@opinheiro.com`, `saremi.saeed@gene.com`

Domain-specific representations limit generalization, as models are not transferable from one modality to another. This narrow focus also limits utility, as most key applications involve interfaces and catalysis across multiple modalities. We argue that modality-agnostic representations are better suited to a wide range of tasks and can learn physical properties across diverse atomic systems, thus leveraging more training data and more challenging metrics. These representations are more expressive [11] as evidenced by the successes of cross-modality models on structure prediction [12, 13], inverse folding [14] or molecular interaction prediction [15].

Here, we introduce *FuncBind*, a unified and scalable framework for generating all-atom molecular systems conditioned on target structures. Following Kirchmeyer et al. [16], we represent molecules with *neural fields*: functions that map 3D coordinates to atomic densities. Neural fields are a compact and scalable alternative to voxels, while sharing many common advantages over point cloud-based representations: they (i) are compatible with expressive neural network architectures (such as CNNs and transformers), (ii) account for variable number of atoms and residues implicitly, and (iii) can represent molecules across modalities with an all-atom formulation. Using this new representation, we build a latent conditional generative model compatible with any score-based approaches. We tested denoising diffusion [17] and walk-jump sampling (WJS) [18], a sampling approach that enjoys fast-mixing and training simplicity when compared to diffusion models, largely because it only relies on one or few noise levels.

We train FuncBind on structures from three drug modalities: small molecules, macrocyclic peptides (MCPs), and antibody complementarity-determining region (CDR) loops in complex with a target protein. These modalities encompass a range of chemical matter with challenging constraints, such as cyclic backbones and non-canonical amino acids. FuncBind achieves competitive results on *in silico* benchmarks, matching or outperforming modality-specific baselines. On *in vitro* experiments, we show that FuncBind can produce novel antibody binders by redesigning the CDR H3 loop of two chosen co-crystal structures. We also create a new dataset$^\dagger$, containing $\sim$190,000 synthetic MCP/protein complexes derived from 641 RCSB PDB structures [19], particularly relevant for this work, as cyclic peptides exhibit chemistry and function that span small and large molecule modalities.

## 2   Related work

**Pocket-conditioned small molecule generation.** Several approaches have framed structure-based drug design as a generative modeling problem [20]. These methods commonly represent molecules—including both ligands and targets—either as point clouds of atoms or as voxel grids. Point-cloud approaches represent atoms as points in 3D space, along with their atomic types, and typically use graph neural network architectures. Point-cloud approaches have been used to generate molecules using autoregressive models [21, 22, 3], iterative sampling approaches [23, 24, 25], normalizing flows [26], diffusion models [27, 4, 28], and Bayesian flow networks [5]. Voxel-based approaches map atomic densities to 3D discrete voxel grids and apply computer vision techniques for generation [29, 6, 30, 31]. VoxBind [7] demonstrates that voxel-based representations achieve state-of-the-art results using expressive vision-based networks and score-based generative models. However, raw voxel-based models do not currently scale to larger molecules due to high memory requirements. Neural fields serve as the continuous analogue to discrete voxels, a technique widely adopted in 3D computer vision [32]. When applied to molecular generation, these fields match existing performance levels while demonstrating superior memory and computational efficiency [16].

**Antigen-conditioned CDR loop generation.** Antibody design is an active research topic and antibody-based treatments represented 26% of 2024 FDA approvals [33] (with related biologics approvals an even higher fraction). A key line of antibody engineering work re-designs the complementarity-determining regions (CDRs), a subset of the heavy and light chains that totals 48 to 82 residues [34] and represents most of the protein's affinity determining variability. A recent approach is to co-design the CDRs sequences and structures using residue cloud representations and equivariant graph neural networks [35, 36, 37, 38], combining these representations with diffusion models for generation [39, 40]. Other works leverage protein language models [41, 42], or revisit the problem by proposing new representations that incorporate domain knowledge and physics-based constraints [43]. As is the case for most models with demonstrated redesign capabilities, we tackle the task where the pose (docking) of the framework is provided. This assumption is relaxed in [44],

---

$^\dagger$available at https://huggingface.co/datasets/Willete3/mcpp_dataset

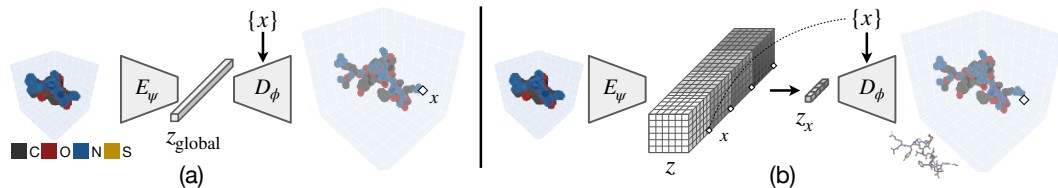

Figure 1: Neural field architectures. (a) Architecture used in [16] where a global embedding is used as input to the neural field decoder. (b) Our proposed neural field architecture, where embeddings are spatially arranged into a feature map grid. The latter allows us to better capture local signal information from input space and is compatible with expressive architectures for denoising.

where the authors finetuned RFDiffusion [45] to generate the positions of backbone CDR atoms, followed by an inverse folding step [10]. Unlike other methods, FuncBind is the first approach based on neural fields that is also applicable to different data modalities simultaneously.

**Pocket-conditioned macrocyclic peptide generation.** Occupying a unique chemical space between small and large molecules, cyclic peptides are a rapidly growing class of therapeutics that can access biological targets often challenging for both small molecules and antibodies [46]. *De novo* generation of target-specific cyclic peptides is therefore highly desirable, in part due to the power and utility of high throughput screens for cyclic and linear peptides and peptoid binders, yet only few works have investigated this. Rettie et al. [47] propose an approach based on RFDiffusion [45] that successfully designed cyclic peptides conditioned on a target protein structure. Yet this prior method is unable to handle non-canonical amino acids, an essential component for both compatibility with industry standard high-throughput screens and for enhancing the therapeutic properties of these peptides. The work by Tang et al. [48] is the only work we are aware of that can handle non-canonical amino acids; this model operates on tokenized smiles and performs target conditioning via classifier guidance with ML-based property predictors that are known to generalize poorly out-of-distribution [49].

## 3 Method

FuncBind is a latent score-based generative model that consists of two training steps: we first learn a latent representation for each molecule that modulates the parameters of a neural field decoder (Section 3.1), then we train a conditional denoiser on these latents (Section 3.2.1). The denoiser is used to sample molecules, conditioned on a given target (Section 3.2.2).

### 3.1 Neural field-based latent representation

We consider a dataset of $N$ molecular complex tuples $\mathcal{D} = \{(v, v^{\text{tar}}, c)_i\}_{i=1}^N$, where $v$ and $v^{\text{tar}}$ are the binder and target, respectively, and $c$ is the modality of the binder. In this work, we focus on three modalities: small molecules, macrocyclic peptides, and CDR loops, though the framework accommodates any atomic system. Atoms are represented as continuous Gaussian-like densities in 3D space and molecules as functions mapping coordinates $x$ to $n$-dimensional atomic occupancy values, $v : \mathbb{R}^3 \to [0,1]^n$ (where $n$ is the number of atom types) [50, 51, 52]. See Section B for details.

Similar to [16], an encoder embeds a molecule into a latent $z$, which is used to decode back an atomic density field. Decoding consists in modulating the parameters of a shared neural field decoder based on the latent representation. However, instead of representing the latent $z$ with a *global* embedding (Figure 1a) used in [16], we consider a *spatially* arranged feature map (Figure 1b). This approach has been successfully applied in other domains [53, 54, 55] and provides two key advantages: (i) each spatial feature captures local information helping to scale the model to larger molecules, (ii) it is compatible with expressive architectures (*e.g.* U-Nets [56]) for denoising.

The encoder $E_\psi : \mathbb{R}^{n \times L^3} \to \mathbb{R}^d, d = C \times L^3$, is a 3D CNN parameterized by $\psi$ that maps a voxel grid $G_v$, generated by discretizing $v$ at a fixed low resolution (for computational efficiency) set by the integer $L$, into a latent space with $C$ channels. For decoding, we use nearest neighbor interpolation as in [54]: from the feature map $z$, we extract position-dependent vectors $z_x \in \mathbb{R}^C$. The embedding $z_x$ is constant over a 3D patch in coordinate space. The decoder $D_\phi : \mathbb{R}^C \times \mathbb{R}^3 \to \mathbb{R}^n$, parameterized

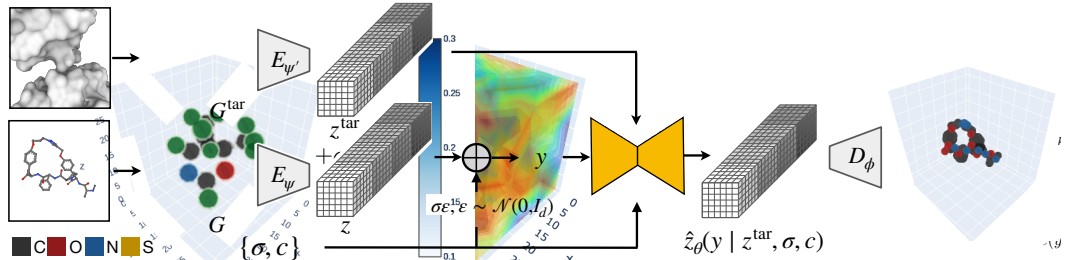

Figure 2: Conditional denoiser training overview. We voxelize separately the binder $v$ and the target $v_{\mathrm{tar}}$ of a given complex and encode them into $z, z^{\mathrm{tar}}$ using encoders $E_\psi, E_{\psi'}$, respectively. We train a denoiser $\hat{z}_\theta(y \mid z^{\mathrm{tar}}, \sigma, c)$ to remove the noise from $y$ conditioned on $z_{\mathrm{tar}}$, the noise level $\sigma$ and the one-hot modality class $c$ (e.g. a cyclic peptide). The denoised latent representation is fed into a neural field decoder $D_\phi$; this gives a reconstructed field $\hat{v}$. $\hat{v}$ undergoes some additional postprocessing to recover the bonds and residue identities (if applicable); see Section 3.3.

by $\phi$, then computes the molecular density field at coordinates $x \in \mathbb{R}^3$, given a local modulation embedding $z_x \in \mathbb{R}^C$. We use a conditional neural field based on multiplicative filter networks [57, 58] with Gabor filters, a natural choice for modeling the sparse atomic density fields [16].

The neural field is trained across modalities with the objective proposed in [16]. A KL-regularization term [59] is added, following common practice in latent generation [60]:

$$\mathcal{L}_{\mathrm{AE}}(\psi, \phi) = \sum_{v \in \mathcal{D}} \mathbb{E}_{z \sim q_\psi(z|v)} \left[ \int \|D_\phi(x, z) - v(x)\|_2^2 \, \mathrm{d}x \right] + \beta \, \mathrm{KL} \left( q_\psi(z \mid v) \mid \mathcal{N}(z; 0, I_d) \right), \quad (1)$$

where $q_\psi(z \mid v) = \mathcal{N}(z; \mu(v), \mathrm{diag}(\sigma(v))^2 I_d)$, $\mu(v)$ and $\sigma(v)$ are parameterized by $E_\psi$ and $\beta$ is a regularization weight. As [16], when optimizing for Equation (1), we upsample coordinates $x$ close to the center of each atom to focus training on non-empty spaces. Since our model does not have equivariance built in the architecture, we apply data augmentation (translation and rotation).

## 3.2 Conditional latent score-based generation

We train a conditional denoiser on the neural-field based representations (Section 3.2.1). The denoiser is used to sample molecules from the aggregate posterior of the VAE encoder [61], conditioned on a given target, with conditional diffusion and walk-jump sampling (Section 3.2.2).

### 3.2.1 Conditioned denoiser

Our denoiser takes as input a noisy latent representation and a set of conditioning information, and outputs the "clean" version of the latent representation. In this work, we condition the denoiser on three signals: (i) the target structure $v^{\mathrm{tar}}$, (ii) the molecule modality $c$ and (iii) the noise level $\sigma$.

More formally, let $(v, v^{\mathrm{tar}}, c)$ be a (binder, target, modality) tuple from the dataset, $(z, z^{\mathrm{tar}}) := (E_\psi(G_v), E_{\psi'}(G_v^{\mathrm{tar}}))$ their latent representations and $y = z + \sigma\varepsilon$, $\varepsilon \sim \mathcal{N}(0, I_d)$, a noisy version of $z$. The target encoder $E_{\psi'}$ is a 3D CNN with similar architecture as $E_\psi$ but different parameters $\psi'$. Following the preconditioning pre-processing proposed by [62], our denoiser $\hat{z}_\theta$ is defined as:

$$\hat{z}_\theta(y \mid z^{\mathrm{tar}}, \sigma, c) = \frac{1}{\sigma^2 + 1} y + \frac{\sigma}{\sqrt{\sigma^2 + 1}} U_\theta \left( \frac{1}{\sqrt{\sigma^2 + 1}} y, z^{\mathrm{tar}}, \frac{1}{4} \log(\sigma), c \right),$$

where $U_\theta$ is a neural network parameterized by $\theta$ and the embeddings $z$ and $z^{\mathrm{tar}}$ are normalized to unit variance and zero mean per channel, similar to [60]. Figure 2 shows an overview of the model architecture. The spatial structure of the latent space allows us to model $U_\theta$ with 3D U-Nets, a standard architecture for generative models in computer vision. In particular, we adapt the network of Karras et al. [62]—designed to generated 2D images—to our 3D generation setting. Crucially, similar to recent works [52, 7, 63], we do not use any type of SE(3) equivariance constraints. Instead, we replace these constraints with data augmentation (rotations and translations) during training.

The conditional denoiser is trained by minimizing the following loss at a given noise level $\sigma$:

$$\mathcal{L}_\sigma(\theta, \psi') = \mathbb{E}_{(v, v^{\mathrm{tar}}, c) \sim \mathcal{D}, \, z \sim q_\psi(z|v), \, \varepsilon \sim \mathcal{N}(0, I_d)} \left[ \left\| \hat{z}_\theta(z + \sigma\varepsilon \mid z^{\mathrm{tar}}, \sigma, c) - z \right\|_2^2 \right], \quad (2)$$

where $z^{\text{tar}} = E_{\psi'}(G_v^{\text{tar}})$ is the encoding of the low-resolution voxel of the target. We apply the reweighting scheme in [62] across noise levels, *i.e.*:

$$\mathcal{L}_{\text{denoiser}}(\theta, \psi') = \mathbb{E}_{\sigma \sim p(\sigma)} \left[ \frac{\sigma^2 + 1}{\sigma^2} \frac{1}{e^{u(\sigma)}} \mathcal{L}_\sigma(\theta, \psi') + u(\sigma) \right],$$

where $\sigma$ is sampled along some pre-determined distribution $p(\sigma)$ (see Section 3.2.2) and $u(\sigma)$ is a one-layer MLP trained with the denoiser. This effectively reweights the loss based on the noise level.

### 3.2.2  Sampling strategies

We experimented with various score-based sampling strategies in the conditional setting, including the SDE formulation of denoising diffusion [64, 65], widely recognized for its state-of-the-art performance in image generation and walk-jump sampling (WJS), based on a probabilistic formulation of least-squares denoising [18]. Diffusion models operate over a continuous range of noise levels in contrast to WJS which considers only one noise level.

These models rely on the Tweedie-Miyasawa formula (TMF) [66, 67], which relates the least-squares denoiser at a noise level $\sigma$ with the score function at the noise level. Given $y = z + \sigma\varepsilon, \varepsilon \sim \mathcal{N}(0, I_d)$, the conditional extension of TMF was derived in [7]. In our notation, it takes the form:

$$\nabla_y \log p(y \mid z^{\text{tar}}, \sigma, c) \approx s_\theta(y \mid z^{\text{tar}}, \sigma, c) := (\hat{z}_\theta(y \mid z^{\text{tar}}, \sigma, c) - y)/\sigma^2, \tag{3}$$

where $\hat{z}_\theta$ is the minimizer of Equation (2); $s_\theta(y \mid z^{\text{tar}}, \sigma, c)$ is the learned conditional score function.

For diffusion, we follow [65] and generate samples by numerically integrating the reverse-time SDE from noise level $\sigma_{\max}$ to $\sigma_{\min}$, approximating the score function with the learned denoiser $\hat{z}_\theta(y \mid z^{\text{tar}}, \sigma, c)$ and TMF. We adopt the variance exploding formulation and the stochastic SDE sampler from EDM [62]. For WJS, we proceed as in Section B.4.2 and report the results on CDR H3 redesign in Section C.1.2.

### 3.3  Recovering molecules from generated atomic-density fields

To recover the underlying molecular structures from sampled latent codes $z$, we employ a post-processing pipeline inspired by [16]. The initial phase determines atom coordinates by identifying local optima in the neural field. This is achieved by first rendering the latent code $z$ into a 0.25Å resolution voxel, then performing peak detection with MaxPooling filters, and finally refining the coordinates through gradient ascent, which takes advantage of the neural field's differentiability. The second phase involves inferring bonds and, when applicable, amino acid identities from the generated point cloud (coordinates and atom types) using OpenBabel software [68]. This yields `.sdf` files for molecules and peptides and `.pdb` files for proteins. A specific approach for identifying non-canonical amino acids, which are not recognized by OpenBabel, is described in Section D.

## 4  Experiments

We test our model on the following *in silico* settings, covering the three modalities discussed above: (i) small molecule generation conditioned on a protein pocket (Section 4.1); (ii) antibody CDR loops redesign conditioned on an epitope (Section 4.2); and (iii) macrocyclic peptides generation conditioned on a protein pocket (Section 4.3). We also performed *in vitro* validation of antibody CDR loops redesign conditioned on an epitope.

For these tasks, the neural field is jointly trained on all three modalities. We train a 5B parameter model across modalities. Samples are generated via conditioning on the target structure. Note that our network is significantly larger than alternative models; we have observed improved performance in the unified setting for larger networks. See Section B for additional model details and Figure 3 for qualitative samples.

We compared our unified model against specialized models trained independently for each modality. Overall, performance parity was observed across most metrics, with the key exception being unique-ness, which was significantly higher in the unified model. See Section C.1.3 for a comparison on CDR H3 inpainting. Further research exploring transfer learning across a broader set of modalities represents an exciting avenue for future work.

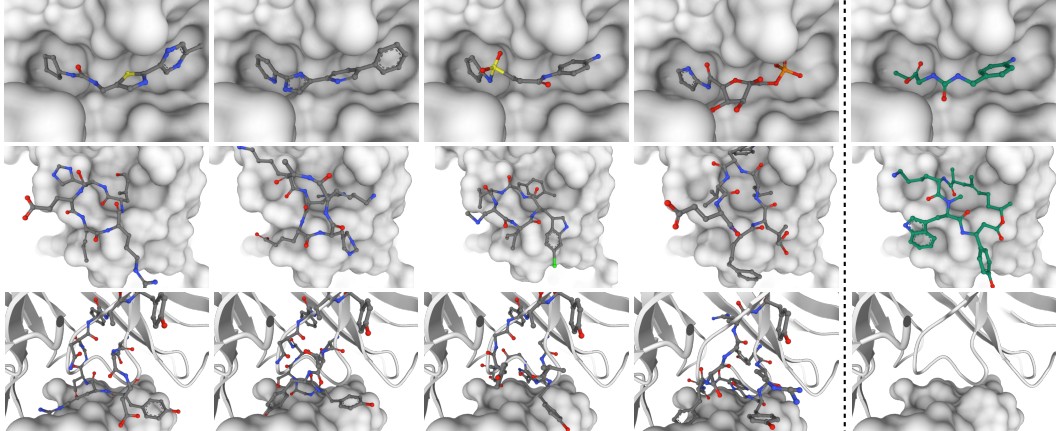

Figure 3: Examples of generated molecules given a target structure for different modalities: (top) small molecules against 2rma, (middle) macrocyclic peptides against 5ooc and (bottom) CDR H3 loop against 5tlk. The seed binders are shown on the right.

## 4.1 Small molecule generation

**Data.** We consider the standard CrossDocked2020 [69] benchmark, with the pre-processing and splitting strategy of [70]. Pockets are clustered at a sequence identity of $< 30\%$ using MMseqs2 and are split into 99,900 train ligand pockets pairs, 100 validation pairs and 100 test pairs.

**Baselines.** We compare FuncBind to various pocket-conditioned ligand generative models: these include point cloud approaches based on autoregressive models (AR [21] and Pocket2Mol [3]), diffusion (DiffSBDD [27], TargetDiff [4], DecompDiff [28]), Bayesian Flow networks (MolCraft [5]) and a voxel-based approach based on walk-jump sampling (VoxBind [7]). FuncBind can be seen as a more scalable generalization of VoxBind and closely matches its performance. All methods but DecompDiff and MolCraft rely OpenBabel [68] to assign bonds from generated atom coordinates.

**Metrics.** We evaluate performance using similar metrics as previous work [4]. For each method, we sample 100 ligands per pocket. We measure affinity with three metrics using AutoDock Vina [71]: *VinaScore* is the docking score of the generated molecule, *VinaMin* is the docking score after a local energy minimization, *VinaDock* fully re-docks the generated molecule, including both search and scoring steps. We also compute the drug-likeness, *QED* [72], and synthesizability, *SA* [73], score of the generated molecules with RDKit [74]. *Diversity* is the average Tanimoto distance (in RDKit fingerprints) per pocket across pairs of generated ligands [75]. *# atoms* is the average number of (heavy) atoms per molecule. We also compute the PoseCheck metrics [76]: *Steric clash* computes the number of clashes between the generated ligands and their pockets, *Strain energy* (SE) measures the difference between the internal energy of the generated molecule's pose (without pocket) and a relaxed pose (computed using Universal Force Field [77] within RDKit); we report the median value.

**Results.** The results are reported in Table 1. FuncBind is competitive with the current state of the art, slightly underperforming on docking-related metrics and strain energy compared to VoxBind and Molcraft and on number of clashes compared to VoxBind. This experiment demonstrates FuncBind's ability to generate highly-variable small molecules. Next, we demonstrate that it can also handle the more regular structures of amino acid-based molecules.

## 4.2 Antibody CDR redesign

**Data.** We consider the SabDab dataset [78], which comprises antibody-protein co-crystal structures and the data splits from DiffAb [39]. This non-i.i.d. split ensures that antibodies similar to those of the test set (*i.e.* more than 50% CDR H3 identity) are removed from the training set. The test split includes 19 targets, for which we redesign each CDR loop individually. As our baselines, we consider the Chothia numbering scheme [79] for the CDR definition.

**Baselines.** We compare FuncBind to representative baselines: RAbD [80], a Rosetta-based method and two ML-based models, DiffAb [39] and AbDiffuser [43]. We consider the variation of AbDiffuser

Table 1: Results on CrossDocked2020 test set. ↑/↓ denote that higher/lower average (Avg.) or median (Med.) is better. For # atoms, numbers close to Reference are better. Baseline results are from [7, 28]. FuncBind's results are shown with mean/standard deviation obtained over 1,000 bootstraps.

| | VinaScore ↓ | | VinaMin ↓ | | VinaDock ↓ | | QED ↑ | SA ↑ | Div. ↑ | S.E. ↓ | Clash ↓ | #atoms |
|---|---|---|---|---|---|---|---|---|---|---|---|---|
| | Avg. | Med. | Avg. | Med. | Avg. | Med. | Avg. | Avg. | Avg. | Med. | Avg. | Avg. |
| *Reference* | -6.36 | -6.46 | -6.71 | -6.49 | -7.45 | -7.26 | .48 | .73 | - | 103 | 4.7 | 22.8 |
| AR | -5.75 | -5.64 | -6.18 | -5.88 | -6.75 | -6.62 | .51 | .63 | .70 | 595 | 4.2 | 17.6 |
| Pocket2mol | -5.14 | -4.70 | -6.42 | -5.82 | -7.15 | -6.79 | .56 | .74 | .69 | 206 | 5.8 | 17.7 |
| DiffSBDD | -1.94 | -4.24 | -5.85 | -5.94 | -7.00 | -6.90 | .48 | .58 | .73 | 1193 | 15.4 | 24.0 |
| TargetDiff | -5.47 | -6.30 | -6.64 | -6.83 | -7.80 | -7.91 | .48 | .58 | .72 | 1243 | 10.8 | 24.2 |
| DecompDiff | -5.67 | -6.04 | -7.04 | -7.09 | -8.39 | -8.43 | .45 | .61 | .68 | N/A | 7.1 | 20.9 |
| MolCraft | -6.59 | -7.04 | -7.27 | -7.26 | -7.92 | -8.01 | .50 | .69 | .72 | 195 | 7.1 | 22.7 |
| VoxBind | -6.94 | -7.11 | -7.54 | -7.55 | -8.30 | -8.41 | .57 | .70 | .73 | 162 | 5.1 | 23.4 |
| FuncBind | -5.71 (±.03) | -5.64 (±.03) | -6.34 (±.03) | -6.18 (±.03) | -7.26 (±.03) | -7.28 (±.03) | .50 (±.002) | .65 (±.001) | .70 (±.0) | 217 (±12) | 7.4 (±.06) | 19.0 (±.09) |

with side chain generation to better match FuncBind's all-atom setting; AbDiffuser in contrast to other baselines, generates all 6 loops jointly. We also compare to AbX [81] and the reproduction of dyMEAN [38] from [81] for H3 design, where the DiffAb splits were considered.

**Metrics.** We compute the following metrics, measuring the similarity of the generated designs to the seed: *amino acid recovery (AAR)*, the sequence identity between the seed and the generated CDRs; *RMSD*, the $C_\alpha$ root-mean-square deviation between the seed and generated structure and *IMP*, the percentage of designs with lower binding energy ($\Delta G$) than the seed, as calculated by `InterfaceAnalyzer` in Rosetta [80]. Baselines apply Rosetta-based relaxations prior to computing IMP to improve the energy scores: DiffAb refines the generated structure with `OpenMM` [82] and AbX uses `fast-relax` [80]. We report metrics for 100 generated samples per target. Note that our model, unlike most baselines, generates samples with diverse sequence lengths; to compute these metrics we consider samples with the same length as the original seed. We found that uniqueness impacts AAR and RMSD, particularly for non-H3 loop designs which exhibited low uniqueness. This presents a challenge for fair model comparison, as baselines do not report uniqueness. For completeness, we also report the metrics for the unique samples on Table 4 (Section C.1).

**Results.** The results in Table 2 indicate that our model is state of the art both on amino acid recovery (AAR) and $C_\alpha$ RMSD values, outperforming other baselines by *1.5 to 3 times* across all loops. This performance can be attributed to our all-atom formulation and neural-field representation, which enable the model to better capture the molecular conformation and conditioning context. AbDiffuser also leverages side-chain information but underperforms in RMSD, highlighting the distinct advantages of our approach. Finally, FuncBind's interface energy improvement (IMP) without backbone minimization is competitive to the IMP of approaches that apply minimization. This showcases the quality and fidelity of the generated structures, as energy is very sensitive to wrong atom placement. As the baselines [81, 39], when applying Rosetta's `fast-relax` backbone minimization on the generated loops, IMP greatly improves as expected, outperforming even the Rosetta RAbD protocol that directly optimizes the energy function. This refinement procedure slightly increases RMSD, as it changes the loop to minimize strain, while FuncBind is trained to mimic patterns in ground-truth crystal structures.

**Length distributions generated.** The above evaluation restricted designs to the seed's length, a common prior in many generative models for this task. However, in many settings, we do not know what is a reasonable length. FuncBind is designed to sample designs across various lengths, a useful capability for *de novo* CDR generation. To demonstrate this flexibility, we analyzed histograms of sequence lengths and atom counts for CDR H3s designed for a de novo target, comparing them against the original seed's values (see Figure 4). For this specific target, while the generated designs exhibited a range of lengths, their distributions were centered on the seed's reference values. Further validation of designs with other lengths is left for future research.

Table 2: CDR inpainting on SAbDab [78] with DiffAb splits [70]. RMSD is in Å and AAR, IMP are in %. † indicates additional relaxation / optimization with Rosetta.

| Method | H1 | | | L1 | | |
|---|---|---|---|---|---|---|
| | AAR↑ | RMSD↓ | IMP↑ | AAR↑ | RMSD↓ | IMP↑ |
| RAbD† | 22.9 | 2.26 | 43.9 | 34.3 | 1.20 | 46.8 |
| DiffAb† | 65.8 | 1.19 | 53.6 | 55.7 | 1.39 | 45.6 |
| AbDiffuser | 76.3 | 1.58 | - | 81.4 | 1.46 | - |
| FuncBind | 86.9 | 0.41 / 0.44† | 35.0 / 77.2† | 86.4 | 0.68 / 0.73† | 45.0 / 80.4† |
| | **H2** | | | **L2** | | |
| RAbD† | 25.5 | 1.64 | 53.5 | 26.3 | 1.77 | 56.9 |
| DiffAb† | 49.3 | 1.08 | 29.8 | 59.3 | 1.37 | 50.0 |
| AbDiffuser | 65.7 | 1.45 | - | 83.2 | 1.40 | - |
| FuncBind | 78.2 | 0.52 / 0.54† | 31.7 / 61.4† | 86.2 | 0.83 / 0.84† | 39.5 / 66.0† |
| | **H3** | | | **L3** | | |
| RAbD† | 22.1 | 2.90 | 23.3 | 20.7 | 1.62 | 55.6 |
| DiffAb† | 26.8 | 3.60 | 23.6 | 46.5 | 1.63 | 47.3 |
| dyMEAN† | 29.3 | 4.80 | 5.26 | - | - | - |
| AbX† | 30.3 | 3.41 | 42.9 | - | - | - |
| AbDiffuser | 34.1 | 3.35 | - | 73.2 | 1.59 | - |
| FuncBind | 47.5 | 2.04 / 2.10† | 19.4 / 49.9† | 80.8 | 0.68 / 0.73† | 32.7 / 67.5† |

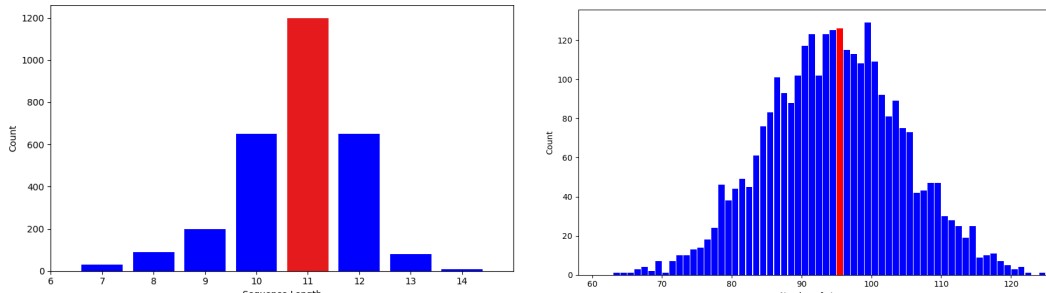

Figure 4: CDR H3 length (left) and atom count (right) histogram on the *de-novo* `4cni` target. Red is the seed H3's reference numbers.

***In vitro*** **evaluation.** We performed wet-lab validation of H3 loop redesigns based on the co-crystal structure of an antibody bound to a rigid and a flexible epitope[‡]. We selected the H3 loop for its important contribution to the antibody's functional properties. We consider a *de novo* setting, where interfaces similar to the two complexes, identified using Ab-Ligity [83], were excluded from training.

From an initial pool of 10,000 unique generated H3 designs (all matching the original seed's length), 190 were selected for experimental testing. This selection involved two steps: 1) The top 500 designs were shortlisted based on model confidence, as indicated by their repeated generation counts. Our *in silico* validation showed that repeats is an useful proxy for high amino acid recovery of the seed. 2) These 500 designs were then clustered into 190 groups using weighted K-means based on sequence edit distance, where the weights were defined by the repeat generation count. The design with the highest repeat count (highest confidence) from each of these 190 clusters was chosen for synthesis and characterization. The selected antibody designs were expressed and purified in the wet lab. Binding affinity was then determined using surface plasmon resonance (SPR) measurements. Section C.2 presents some detailed analysis. FuncBind achieves a binding rate of 45% on the rigid epitope and 2% on the flexible epitope, which increases to 4% with a relaxed binding threshold.

---

[‡]For legal reasons, we do not disclose the target's names.

Table 3: Results on our MCP benchmark. RMSD is in Å, Residues-TS≥0.5, Vina Dock are in %.

| | TS↑ | Residues-TS≥0.5↑ | L-RMSD↓ | I-RMSD↓ | TM-Score↑ | Vina dock↑ |
|---|---|---|---|---|---|---|
| RFPeptide | 0.31 | 29 | 12 | 3.3 | 0.33 | 8.8 |
| AfCycDesign | 0.34 | 29 | 7.6 | 3.7 | 0.33 | 29 |
| FuncBind | 0.33 | 25 | 2.6 | 1.8 | 0.36 | 41 |

## 4.3 Macrocyclic peptide generation

**Data.** Given the scarcity of established baselines, benchmarks, and available data for macrocyclic peptide (MCP)-protein complexes, we introduce a novel benchmark to facilitate the evaluation of generative models for MCPs. To address the data limitation, we have curated a dataset of 186,685 MCP-protein complexes using a "mutate then relax" strategy detailed in Section E. Taking as input an original set of 641 protein-MCP complexes sourced from RCSB PDB [19], this strategy consists in (i) randomly mutating the MCPs at 1 to 8 different sites, using a list of 213 distinct amino acids, (ii) relaxing them using `fast-relax`, which involves iterative cycles of side-chain packing and all-atom minimization [84] and (iii) selecting the lowest interface scores. The source dataset comprises lengths ranging from 4 to 25 amino acids with an average of 10 (Section E Figure 13a). 78% of the MCPs contain one or more non-canonical amino acids, *i.e.* any amino acid that is neither L-canonical nor D-canonical. We split the dataset into train, test and validation subsets using a clustering approach detailed in Section E that aims at creating a non-i.i.d. test set consisting of 85 protein pockets.

**Baselines.** MCPs pose significant challenges for generative models due to their non-canonical amino acids, cyclization, and scarce training data. To our knowledge, no other target-conditioned, structure-based MCP generative models handles non-canonical amino acids, precluding direct comparisons. For reference, we compare nonetheless FuncBind with AfCycDesign [85] and RFPeptide [47], two models generating MCPs exclusively with canonical amino acids and N-to-C cyclization.

**Metrics.** As part of this new benchmark, we define and compute relevant metrics. *Tanimoto similarity (TS)* assesses the resemblance between the ground-truth seed and the sampled MCP structure. *Ligand RMSD (L-RMSD)* is the RMSD between sample MCP to seed MCP, and *template modeling (TM) score*, a length independent similarity metric, based on the Kabsch alignment of the backbone atoms $(N, C_\alpha, C, O)$. TM score was calculated by maximizing the scaling factor. The same backbone logic is applied to compute *interface RMSD (I-RMSD)*, the RMSD in the pocket (which are for the most part slightly lower since the pockets are identical). The sample and the seed were not aligned for I-RMSD since this is based on where the MCPs are in the pocket. Binding affinity was calculated through *Autodock Vina* [71].

**Results.** We observe a correlation between the generated designs with the MCP seeds. Qualitatively, Section E.1 Figure 10 illustrates the close alignment of the backbone between the sampled structures and the seed. Most of the sampled molecules display consistent repeating peptide bonds, linking the $C_1$ carbon of one $\alpha$-amino acid to the $N_2$ nitrogen of the next. Closure bonds (such as disulfide in Figure 10a,b and N to C cyclization in Figure 10c) are also often maintained in the sampled sets.

Metrics are reported in Table 3. The mid-range TS and TM scores reflects a strong similarity to the peptide backbone, with variability occurring at the functional groups of the residues. An example of per-residue TS for molecules sampled with the seed mutant (Section E.1 Table 10a) and the crystal MCP (Section E.1 Table 10b) shows that the highest TS occurs at the disulfide closure bond. This elevated per-residue similarity results from the preservation of closure bond residues throughout the curated dataset. Furthermore, the per-residue TS is higher across the crystal MCP residues than at the mutated residues of the seed mutant (PRO3A20 and GLU4B60). Because all mutants originate from the crystal MCP, the dataset is closely tied to the crystal sequence, and the sampled structures similarly reflect this connection. Low RMSD results show generally good alignment with the seed MCP, with many samples exhibiting RMSDs below 1Å, particularly for I-RMSD. These results are consistent with the TM scores, where ∼20% of the samples exhibit TM scores greater than 0.5. Finally, Autodock Vina binding affinity reveals that nearly half of the generated samples, both before and after minimization, have better binding affinity in the pocket compared to the seed.

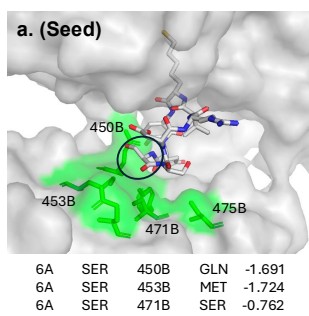 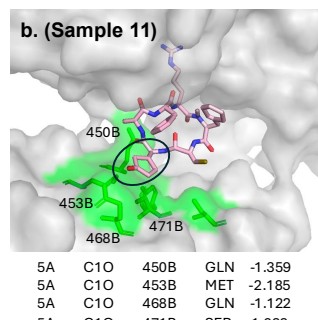 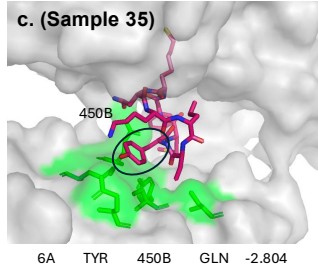

| 6A | SER | 450B | GLN | -1.691 |
| 6A | SER | 453B | MET | -1.724 |
| 6A | SER | 471B | SER | -0.762 |

| 5A | C1O | 450B | GLN | -1.359 |
| 5A | C1O | 453B | MET | -2.185 |
| 5A | C1O | 468B | GLN | -1.122 |
| 5A | C1O | 471B | SER | -1.633 |

| 6A | TYR | 450B | GLN | -2.804 |

Figure 5: Per-residue energy scores at the same position were calculated using Rosetta's residue energy breakdown for a seed and two samples. We analyzed: (a) the seed's serine, (b) 3-hydroxycyclopentyl-alanine (C1O) from sample 11 (Section E.1 Figure 12), (c) tyrosine from sample 35.

Compared to the baselines, FuncBind achieves superior or similar metrics, notably lower L/I-RMSDs. FuncBind also yields the highest proportion of designs with better docking scores. The only underperformance was in Tanimoto similarity (Residues-TS≥0.5 in particular), expected as FuncBind accesses a larger set of non-canonical amino acids and sequence lengths as opposed to these baselines. We encourage future comparisons on this new benchmark, especially for models handling non-canonical amino acids.

**Analysis of generated non-canonicals.** In Figure 11 (Section E.1), amino acids are categorized into known canonical and non-canonical amino acids (seen in the training set), and unknown non-canonical amino acids, which represent newly generated amino acids not previously seen. Fewer than 1% of all categorized amino acids were labeled as "unreasonable", a designation applied when a bond was shorter than 0.8Å or when invalid oxygen–oxygen or nitrogen–nitrogen bonds were present. Some reasonable and novel generated amino acids are presented in Figure 12 (Section E.1). Generating novel, chemically plausible amino acids without restrictions from a predefined library or initial cyclic backbone allows broader exploration of the binding pocket. This is demonstrated in Figure 5b, where an amino acid absent from our library interacts with pocket residues that neither the seed (Figure 5a) nor a chemically similar amino acid at the same position (Figure 5c) engage. The absence of constraints in MCP generation promotes greater sequence diversity and deeper investigation of the binding pocket.

# 5   Conclusion

We presented FuncBind, a new framework for all-atom, structure-conditioned *de novo* molecular design. FuncBind is based on a new modality-agnostic representation, that enables a single model to be trained across diverse drug modalities; we focused on small molecules, macrocyclic peptides and antibody CDRs. FuncBind handles variable atom and residue counts and is based on recent advancements in computer vision, replacing equivariance constraints with data augmentation. FuncBind demonstrates competitive *in silico* performance, matching or outperforming specialized baselines. *In vitro*, we demonstrate that FuncBind generates binders against *de novo* targets. It generates novel and chemically plausible molecules, including new non canonical amino acids. Future directions include extending FuncBind to larger biomolecular systems and to more data modalities. Furthermore, the scaling behaviour of this model remains an interesting direction for future study, particularly given the absence of overfitting as the denoiser increased in size (we tested up to 5B parameters).

It is important to note that, like other structure-based methods, FuncBind relies on the availability of an accurate model of the molecular interface to be designed. This can be a limitation, as these models are costly to obtain and their availability is often restricted in the drug discovery process, particularly for large molecules. Finally, real-world application of generative models in drug design requires addressing a range of properties beyond binding, *e.g.* synthesizability for small molecules and developability for antibodies, considerations not handled in this work.

**Acknowledgements**   We thank Prescient Design and the following colleagues: Jan Ludwiczak for processing the SabDab dataset. Tamica D'Souza for performing the antibody wetlab validation.

Max Shen, Namuk Park, Nathan Frey, Sidney Lisanza, Rob Alberstein, Ewa Nowara, Natasa Tagasovska, Chen Cheng, Pan Kessel, Sarah Robinson, Joshua Yao-Yu Lin for insightful discussions and Genentech's legal team.

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

# Appendices

This supplementary material is organized as follows:

## A    Broader impacts

This work introduces FuncBind, a novel framework for all-atom, structure-conditioned molecular generation, whose primary positive impact lies in its potential to accelerate and enhance the discovery of new therapeutics across diverse modalities, from small molecules to complex biologics like peptides and antibodies. While the underlying principles could find applications in other scientific fields like materials science, its deployment in drug discovery requires addressing significant challenges, including the validation gap between in silico predictions and experimental testing (in vitro, in vivo, and clinical trials).

## B    Model details

### B.1    Representation

FuncBind is based on a neural field representation that models an atomic density field, a smooth function taking values between 0 (far away from all atoms) and 1 (at the center of atoms). This field takes the following form [50, 51]:

$$\forall x \in \mathbb{R}^3,\ v_a(x) = 1 - \prod_{i=1}^{n_a} \left( 1 - \exp\left( - \left( \frac{\|x - x_{a_i}\|}{.93r} \right)^2 \right) \right), \tag{4}$$

where $a_i$ is the $i^{\text{th}}$ atom of type $a$ (among $n$ choices), for a total of $n_a$ atoms and $r$ is the atoms' radius set to $r = 1.0\text{Å}$ for all atom types.

We consider $n = 8$ element types $C, O, N, S, F, Cl, P, Br$ that cover all major atom types across small molecule, macrocyclic peptides and proteins. Note that protein-specific atom types (e.g. $C_\alpha, C_\beta$ etc.) are merged into a single element type (e.g. $C$). This helps transfer learning across modalities.

Finally, the field is defined over a volume of $(32\text{Å})^3$. It is the continuous version of a voxel grid of spatial dimension 128 and resolution of 0.25Å, in $\mathbb{R}^{8 \times 128^3}$.

### B.2    Neural Field

The encoder $E_\psi$ is a 3D CNN containing 4 residual blocks (number of hidden units 256, 512, 1024, 2048 for each block), where each block contains 3 convolutional layers followed by BatchNorm, ReLU and pooling layers (we use max pooling on the first three blocks). The encoder has 59M parameters. The input to the encoder is a low-resolution grid of spatial grid dimension $L = 16$ corresponding to a resolution of 2Å. Before voxelizing the molecules, we first center the atoms around the tightest bounding box encapsulating the molecule, apply a random rotation to the atoms

(each Euler angle rotated randomly between $[0, 2\pi)$) and random translation between $[-1, 1]$Å then normalize their coordinates to the range of $[-1, 1]$.

The decoder $D_\phi$ is a conditional Multiplicative Filter Network (MFN) [57, 58] with Gabor filters and 6 FiLM-modulated layers, where each fully-connected layer has 2048 hidden units. The decoder has 59M parameters.

The auto-encoder is trained with Adam Optimizer [86] with learning rate $10^{-2}$, $\beta_1 = 0.9$, $\beta_2 = 0.999$. We apply a KL regularization weight of $\lambda = 10^{-5}$. Batch size is 32 over 1 B200 GPU; we sample 15000 coordinates per batch.

### B.3 Denoiser

The hyperparameters of the Karras et al. [62] UNet architecture are as follows: 5B model with 8 blocks with 512 channels, channel multipliers [1,2,3,4], attention resolutions [4, 2]. This model follows the XXL setting of EDM2 [62] and increased the number of channels from 448 to 512. For reference, Protéina [63], the largest protein backbone generative model, comprises 400M parameters and RFDiffusion [8] roughly 100M parameters. As [65], we apply preconditioning to learn the denoiser across noise levels. Moreover, we sample the noise levels from a log-normal distribution with mean 1.2 and standard deviation 0.8.

The target encoder $E_{\psi'}$ is a 3D CNN which takes as input a voxel grid of dimension $\mathbb{R}^{4 \times 32^3}$ of the target protein (resolution 1.0Å), considering atom elements $C, O, N, S$, and consists of a magnitude preserving [62] CNN layer with kernel $3 \times 3 \times 3$ and output channels 64, then a downsampling layer to a spatial grid in $\mathbb{R}^{64 \times 16^3}$ then a magnitude preserving U-Net block [62] with output channels $C = 128$ leading to a voxel of size $\mathbb{R}^{C \times 16^3}$.

The parameters are optimized with Adam optimizer [86] with learning rate $\alpha_{\text{ref}} = 10^{-2}$, $\beta_1 = 0.9$, $\beta_2 = 0.95$ using an aggregated batch size of 768 over 8 B200 GPUs. We perform early stopping on the validation loss. We use the power function exponential moving average from EDM2 [62] with an EMA length of 5%. Moreover, we adopt the inverse square root decay schedule of [86], also used in [62] which sets $\alpha(t) = \dfrac{\alpha_{\text{ref}}}{\sqrt{\max(t/t_{\text{ref}}, 1)}}$, where we set $t_{\text{ref}} = 20040$. Finally, the networks are trained by randomly dropping the conditioning information 10% of the time.

### B.4 Sampling

To improve uniqueness, we apply different rotations to the pocket on each MCMC chain, in a similar fashion to how rotation-based data augmentation is performed at training time.

#### B.4.1 Denoising diffusion

We set as follows the sampling parameters of EDM2 [62]:

- $N = 128$ steps
- $\sigma_{\min} = 0.01$, $\sigma_{\max} = 10$
- $S_{\min} = 5.0$, $S_{\max} = 7.0$
- $S_{\text{churn}} = 30.0$
- $S_{\text{noise}} = 1.003$
- $\rho = 7$

Moreover, we apply a temperature scaling with $\tau = 0.5$ on Crossdocked and $\tau = 0.33$ on MCP-protein complexes.

#### B.4.2 Walk-Jump sampling

We also implemented a conditional form of the Walk-Jump Sampling (WJS), a score-based generative model that is based on a probabilistic formulation of least-squares denoising [18]. The framework is based on the Tweedie-Miyasawa formula (TMF) [66, 67], which relates the least-squares denoiser

at a noise level $\sigma$ with the score function at the noise level. Given $y = z + \sigma\varepsilon, \varepsilon \sim \mathcal{N}(0, I_d)$, the conditional extension of TMF was derived in [7]. In our notation, it takes the form:

$$\nabla_y \log p(y \mid z^{\text{tar}}, \sigma, c) \approx s_\theta(y \mid z^{\text{tar}}, \sigma, c) := (\hat{z}_\theta(y \mid z^{\text{tar}}, \sigma, c) - y)/\sigma^2, \tag{5}$$

where $\hat{z}_\theta$ is the minimizer of Equation (2); $s_\theta(y \mid z^{\text{tar}}, \sigma, c)$ is the learned conditional score function.

**Walk-jump sampling.** WJS [18] is composed of two stages: (i) *(walk)* samples the noisy latent variables conditioned on $z^{\text{tar}}, c$ using Langevin Markov chain Monte Carlo (MCMC) via the learned score function (Equation (5)), (ii) *(jump)* estimates "clean" $z$ by single-step denoising. There is a fundamental trade-off in this sampling strategy: for larger $\sigma$, sampling from the smoother density becomes easier, but the denoised samples move farther away from the distribution of interest [87].

**Multimeasurement walk-jump sampling.** The sampling trade-off in WJS is addressed in multi-measurement denoising models [88, 87], in which the problem is framed as sampling from the distribution $p_\sigma(y_{1:m})$ associated with $y_{1:m} := (y_1, \ldots, y_m)$, where $y_k = z + \sigma\varepsilon_k$, $k \in \{1, \ldots, m\}$, and $\varepsilon_k \sim \mathcal{N}(0, I_d)$ all independent of $z$. Saremi et al. [87] studied a sequential scheme for sampling from $p_\sigma(y_{1:m})$ and showed that the noise level effectively decreases (as far as the denoiser is concerned) at the rate $\sigma/\sqrt{m}$. Furthermore, it was shown that sampling becomes easier upon accumulation of measurements. The general sampling problem is therefore mapped to a sequence of sampling noisy data at a *fixed* noise scale, while the effective noise decreases via accumulation of measurements. We refer to this scheme as WJS-$m$, which involves two hyperparameters: the noise level $\sigma$, and the number of measurements $m$. In this construction, we only need to keep track of the empirical mean of noisy samples. In particular, we have (see [87, Eq. 4.9]):

$$\nabla_{y_m} \log p_\theta(y_m \mid y_{1:m-1}, z^{\text{tar}}, \sigma, c) = \frac{1}{m} s_\theta(\overline{y}_{1:m} \mid z^{\text{tar}}, \frac{\sigma}{\sqrt{m}}, c) + \frac{1}{\sigma^2}(\overline{y}_{1:m} - y_m),$$

where $\overline{y}_{1:m}$ is the empirical mean of the measurements $(y_1, \ldots, y_m)$. The score function above is used in sampling $(y_1, \ldots, y_m)$ iteratively using Langevin MCMC [87, Algorithm 1]. Finally, the denoising "jump" in WJS-$m$ is achieved via (single-measurement) TMF using the sufficient statistics $\overline{y}_{1:m}$ at the noise scale $\sigma/\sqrt{m}$. It is clear that the vanilla WJS discussed above reduces to WJS-1. Although there is a flavor of diffusion in this scheme due to its sequential strategy, WJS-$m$ is arguably more "surgical" in that, by construction, we do not need to learn score functions over a continuum of noise levels, but only a finite one identified by $m$. This is especially appealing for applications where WJS-1 already shows reasonable performance and $m$ is therefore taken to be small. This work contains the first experimental validation of WJS-$m$ in generative modeling applications.

We report the results for WJS for CDR H3 inpainting in Section C.1.2. The parameters are set to $\sigma = 7.0$ and $m = 16$. We use underdamped Langevin MCMC from Sachs *et al.* [89] in the BAOAB scheme with $K = 50$ steps, friction $\gamma = 1.0$, discretization step $\delta = \sigma/2$ ([87, Algorithm 1]).

# C    CDR redesign

## C.1    In silico evaluation

### C.1.1    Uniqueness

Table 4 reports our CDR sampling results over unique sequences.

We observe that higher uniqueness usually leads to lower Amino Acid Recovery (AAR). In other words, repeated sequences tend to correlate more with the seed. We use this simple heuristic to select H3 designs for in vitro evaluation (see Section C.2).

### C.1.2    Ablation with Walk Jump Sampling

We report the performance of multimeasurement WJS and diffusion. Overall the models are comparable with slightly higher uniqueness for diffusion.

### C.1.3    Comparison to specialized model

We observe that the unified model has higher uniqueness than the specialized model, with slightly better CDR loop inpainting performance on unique samples. We observe this trend on the other data modalities as well.

Table 4: Impact of uniqueness on CDR inpainting performance. RMSD is in Å and AAR and Uniqueness are in %. $\star$ indicates designs with *unique* sequences.

| Method | AAR↑ | RMSD↓ | Unique↑ | AAR↑ | RMSD↓ | Unique↑ |
|---|---|---|---|---|---|---|
| | | H1 | | | L1 | |
| AbDiffuser | 76.3 | 1.58 | - | 81.4 | 1.46 | - |
| FuncBind | 86.9 | 0.41 | 23.5 | 86.4 | 0.68 | 38.9 |
| FuncBind$^\star$ | 75.9 | 0.45 | 100 | 79.3 | 0.85 | 100 |
| | | H2 | | | L2 | |
| AbDiffuser | 65.7 | 1.45 | - | 83.2 | 1.40 | - |
| FuncBind | 78.2 | 0.52 | 20.6 | 86.2 | 0.83 | 19.4 |
| FuncBind$^\star$ | 59.5 | 0.57 | 100 | 54.4 | 2.39 | 100 |
| | | H3 | | | L3 | |
| AbDiffuser | 34.1 | 3.35 | - | 73.2 | 1.59 | - |
| FuncBind | 47.5 | 2.04 | 85.5 | 80.8 | 0.68 | 44.1 |
| FuncBind$^\star$ | 44.1 | 2.16 | 100 | 68.9 | 0.94 | 100 |

Table 5: Diffusion vs WJS on H3 loop inpainting. $\star$ indicates designs with *unique* sequences.

| Method | AAR↑ | RMSD↓ | Unique↑ |
|---|---|---|---|
| FuncBind$_{\text{diff}}$ | 47.5 | 2.04 | 85.5 |
| FuncBind$^\star_{\text{diff}}$ | 44.1 | 2.16 | 100 |
| FuncBind$_{\text{WJS}-16}$ | 51.0 | 1.89 | 73.8 |
| FuncBind$^\star_{\text{WJS}-16}$ | 41.4 | 2.18 | 100 |

Table 6: Loop uniqueness comparison between Unified and Specialized models.

| Loop | Uniqueness Unified | Uniqueness Specialized |
|---|---|---|
| H1 | 23.5 | 9.6 |
| H2 | 20.6 | 12.6 |
| H3 | 85.5 | 69.2 |
| L1 | 38.9 | 10.6 |
| L2 | 19.4 | 14.0 |
| L3 | 44.1 | 22.3 |

Table 7: H3 loop performance on unique samples.

| H3 loop | AAR | RMSD |
|---|---|---|
| Unified | 0.441 | 2.16 |
| Specialized | 0.406 | 2.07 |

## C.2 In vitro validation

To experimentally validate FuncBind's capabilities, we performed wet-lab validation of H3 loop redesigns based on the co-crystal structure of an antibody bound to a rigid and a flexible epitope. We selected H3 loop redesign due to H3's important contribution to the antibody's functional properties. This study was conducted in a de novo setting: interfaces similar to the two complexes, identified using Ab-Ligity [83], were excluded from training.

From an initial pool of 10,000 unique generated H3 designs (all matching the original seed's length), 190 (*i.e.* 2 SPR plates) were selected for experimental testing. This selection involved two steps:

1. The top 500 designs were shortlisted based on model confidence, as indicated by their repeated generation counts. Our in-silico validation showed that repeats is an useful proxy for high amino acid recovery of the seed.

2. These 500 designs were then clustered into 190 groups using weighted K-means based on sequence edit distance, where the weights were defined by the repeat generation count. The design with the highest repeat count (highest confidence) from each of these 190 clusters was chosen for synthesis and characterization.

The selected antibody designs were expressed and purified in the wet lab. Binding affinity was then determined using surface plasmon resonance (SPR) measurements.

### C.2.1 Rigid epitope

An analysis of Amino Acid Recovery (AAR) and Root Mean Square Deviation (RMSD) for the selected designs are provided in Figure 6:

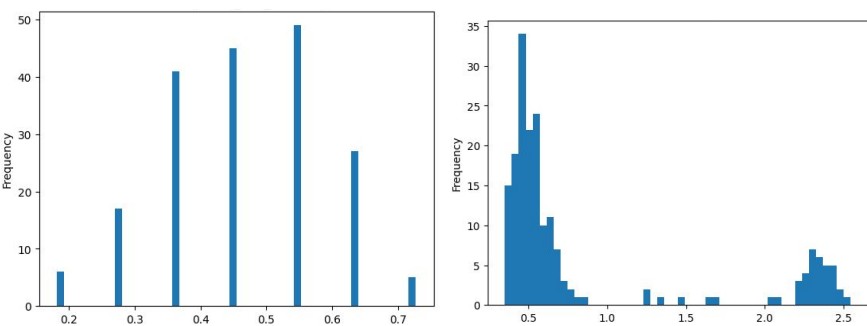

Figure 6: All generated CDR H3 designs on rigid epitope: AAR (left) and RMSD (right) histogram.

FuncBind successfully generated novel antibody binders in this de novo redesign problem; in fact 54% were binders (42% with pKD values). 94% of all 190 submitted designs were successfully expressed and purified. Experimental results confirmed that 42% of all submitted designs were binders, with pKD values in the range of [7.55, 11.29] ($K_D \in [5.08 \times 10^{-12}, 2.84 \times 10^{-8}]M$) and an average pKD of 9.56 ($K_D = 2.00 \times 10^{-9}M$). For comparison, the pKD of the parent antibody is 10.20 ($K_D = 2.63 \times 10^{-11}M$). 12% were binders with no pKD ("bad" binders). We identified a 5X binder in that set.

Table 8: Average RMSD and AAR for binders and non binders on rigid epitope

|  | Binders | Unassigned | Non-Binders | Global |
|---|---|---|---|---|
| RMSD | 0.50 | 0.53 | 1.31 | 0.88 |
| AAR | 55.1 | 50.8 | 37.9 | 46.7 |

Table 9: Average RMSD and AAR for binders and non binders on flexible epitope

|  | Binders | Unassigned | Non-Binders | Global |
|---|---|---|---|---|
| RMSD | 1.93 | 2.08 | 1.94 | 1.95 |
| AAR | 40.4 | 35.6 | 34.2 | 34.5 |

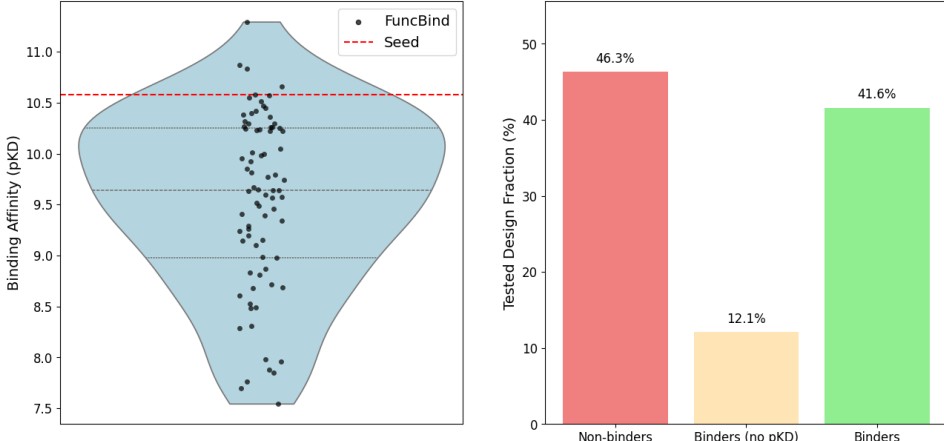

Figure 7: In vitro validation of FuncBind's designs against a rigid epitope; expression (left), binding affinity (center), binding rate (right). Expression rate is 93.68%.

## C.3 Flexible epitope

An analysis of Amino Acid Recovery (AAR) and Root Mean Square Deviation (RMSD) for the selected designs are provided in Figure 8:

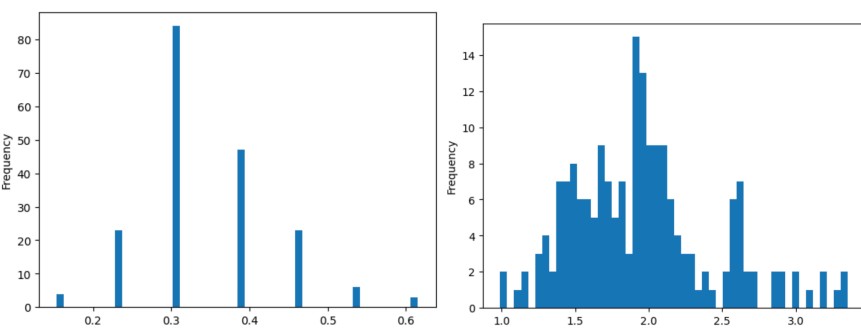

Figure 8: All generated CDR H3 designs against a flexible epitope; target: AAR (top left) and RMSD (top right) histogram. Bottom: Logo of generated designs.

Experimental results confirmed that 100% of the designs expressed and 2% of all submitted designs were binders, with pKDs [6.95, 7.44, 8.08, 8.47] ($K_D \in [3.38 \times 10^{-9}, 1.13 \times 10^{-7}]M$) and an average pKD of 7.74 ($K_D = 4.03 \times 10^{-8}M$). For comparison, the pKD of the parent antibody is 10.58 ($K_D = 6.32 \times 10^{-11}M$). 10% were binders with no pKD ("bad" binders); around 3 Unassigned binders had a very reasonable SPR curves.

Looking at Table 9, no specific correlation between binders and non binders were found based on RMSD on the limited set of binders we had. Though higher AAR seemed to be better (based on 4 binder samples only).

Table 10: Per-residue TS between the (a) seed MCP and (b) crystal MCP of the 20 sampled molecules in the pocket relaxed to 1vwe mutant 90.

| (a) | CYS | HIS | A20 | B67 | PHE | CYS |
|---|---|---|---|---|---|---|
| 1vwe_mut90 | $0.63 \pm 0.24$ | $0.24 \pm 0.17$ | $0.26 \pm 0.07$ | $0.14 \pm 0.12$ | $0.29 \pm 0.14$ | $0.53 \pm 0.21$ |

| (b) | CYS | HIS | PRO | GLU | PHE | CYS |
|---|---|---|---|---|---|---|
| 1vwe-CP | $0.61 \pm 0.23$ | $0.24 \pm 0.12$ | $0.31 \pm 0.14$ | $0.28 \pm 0.10$ | $0.33 \pm 0.08$ | $0.57 \pm 0.23$ |

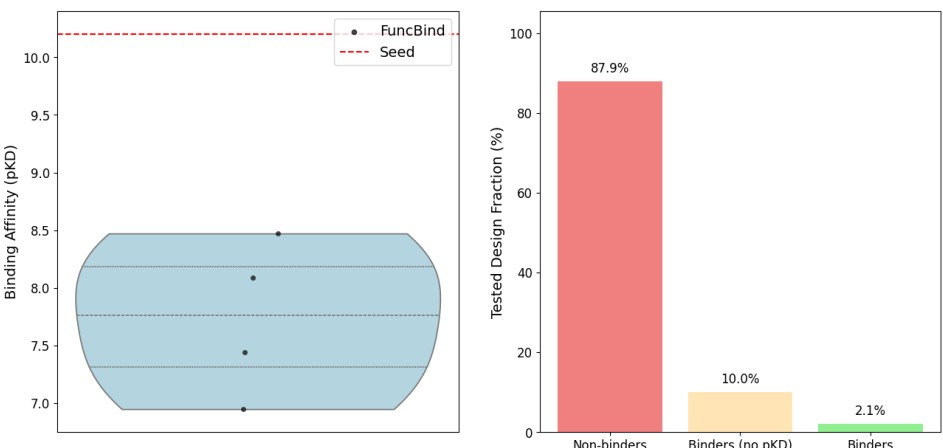

Figure 9: In vitro validation of FuncBind's designs against a flexible epitope; expression (left), binding affinity (center), binding rate (right). Expression rate is 100%.

## D    Identifying non canonical amino acids

Unidentified amino acids were determined by recognizing repeated patterns of peptide backbone atoms around a chiral carbon. All atoms stemming from a $C_\alpha$, including those in the side chain, were identified and labeled per canonical atom naming conventions. SMILES strings of unidentified amino acids were compared with a non-canonical amino acid library, assigning residue names when a match was found. If no match was found, the amino acid was labeled as unknown and added to the non-canonical library. In the output PDB file, a canonical residue name was assigned based on the closest alignment of atom naming patterns (*e.g.*, $C_\gamma$, $C_{\delta_1}$, $N_{\epsilon_1}$) to a known canonical residue.

## E    Macro cyclic peptides

### E.1    Additional results

We perform the following studies:

- Figure 10: qualitative comparison between some MCP samples and the seed.

- Table 10: per residue tanimoto similarity (TS) for molecules samples with the seed mutant (a) and the crystal MCP (b).

- Figure 11 shows the categorization of the generated amino acids.

- Figure 12 shows some reasonable and novel generated non canonical amino acids.

- Figure 5 shows how a newly generated amino acid interacts with pocket residues that neither the seed nor a chemically similar amino acid at the same position engage.

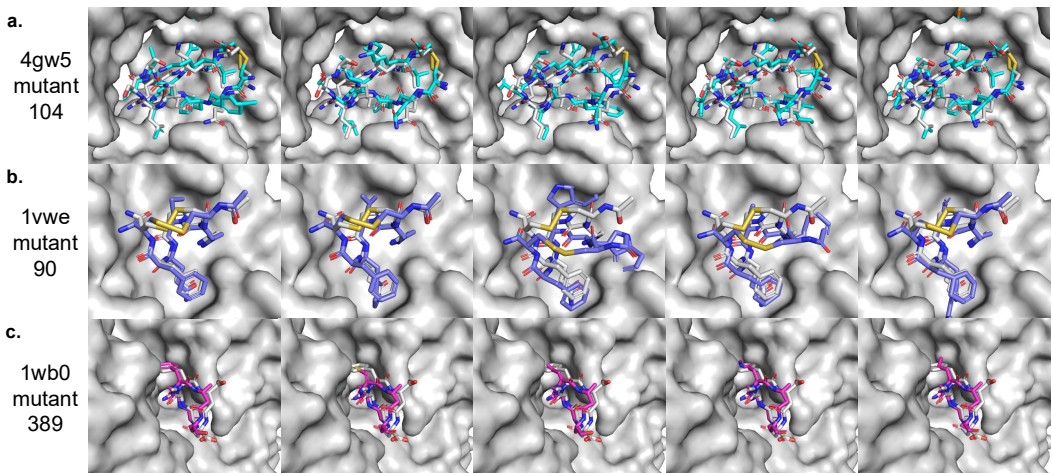

Figure 10: Results of sampling in pocket relaxed around MCP seed (grey) are in (a) blue in 4gw5 mutant 104 pocket, (b) purple for 1vwe mutant 90 pocket, and (c) 1wb0 mutant 389 pocket

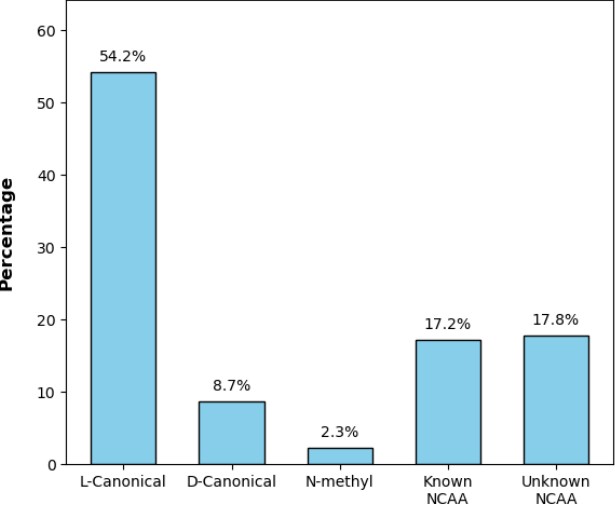

Figure 11: Proportion of amino acid types classified as L-canonical, D-canonical, N-methylated, other known non-canonical amino acids (as annotated in our library), and unknown non-canonical amino acids in the sampled set.

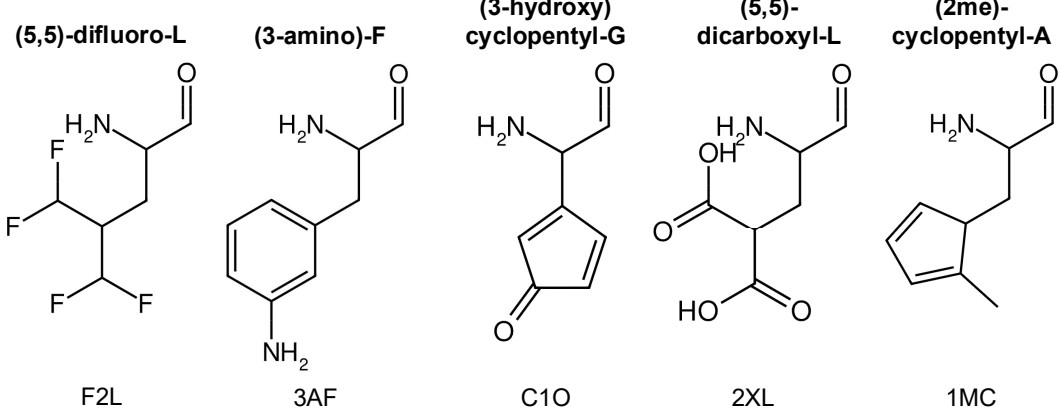

Figure 12: Examples of unknown or novel NCAAs that appeared in the sampled set but were not present in the initial test set library. Novel NCAAs are labeled with a formal name and an assigned 3-letter AA code.

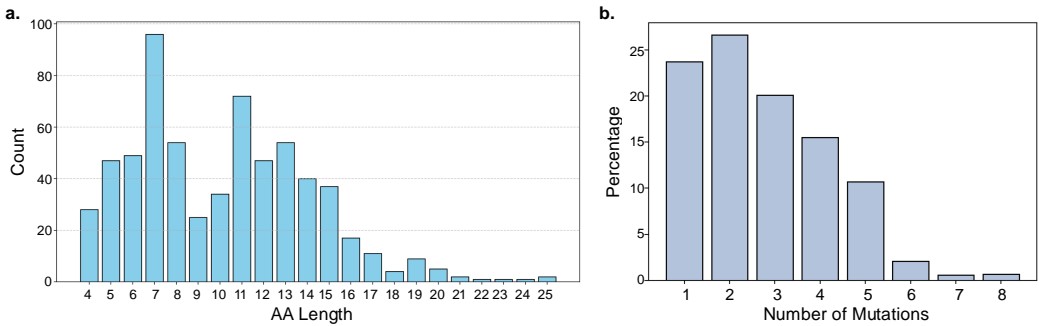

Figure 13: (a) Count of MCPs with each amino acid length in the source MCP-protein dataset. (b) Percentage of the number of mutations in the curated MCP-protein bound dataset.

## E.2 Data curation

The MCP-protein pair dataset was curated by randomly mutating the MCPs from the source dataset at 1–8 different sites, using a list of 213 distinct amino acids (Figure 13b). The closure bonds in the source dataset consist of 55% N-to-C (head-to-tail), 28% disulfide (cysteine-cysteine), and 23% S-acetyl-cysteine. The remaining 4% contain other closure bonds, such as linkers, and were mainly avoided or modified in the curated MCP dataset so that mutations can be easily implemented in Rosetta. Mutations were avoided in amino acids involved in disulfide bonds and S-acetyl-cysteine linkage. The amino acid list used for random mutation of the MCPs included L-canonical, D-canonical, N-methylated, and other non-canonical types—such as alpha-modified, beta-modified, and peptoid amino acids. Many of these non-canonical residues were pre-parameterized and available in the Rosetta non-canonical rotamer libraries [90]. Following mutation, the MCPs were relaxed using the `fast-relax` protocol, which involves iterative cycles of side-chain packing and all-atom minimization [84]. The Rosetta interface energy scores—representing the binding energy of the protein-peptide complex at each position—were calculated using the `ref_2015_cart` energy function. From each source MCP-protein structure, over 2,000 mutated and relaxed complexes were generated, and approximately 500 with the lowest Rosetta interface scores were selected for the curated dataset. Therefore, we were able to expand the source dataset to 186,685 total MCP-protein structures in the curated dataset.

## E.3 Clustering and Splitting

The pairwise similarity of protein sequences from the 641 protein targets in the curated dataset was evaluated using the Longest Common Subsequence (LCS) method [91]. Similarity between each

pair of sequences was calculated as the ratio of the length of their LCS to the length of the longer sequence. The target proteins were clustered together under a representative if their similarity score was greater than 0.5. If none of the similarity scores met the 0.5 threshold, a new cluster was created with that protein as the representative. 208 distinct protein clusters were used for training, validation, and testing. Clusters containing more than 100 MCP-protein pair structures in total were included in the training set. From the remaining clusters, 100 were randomly assigned to the test set, while the rest were added to the validation set.

