# OpenReview forum: "Unified all-atom molecule generation with neural fields"
_NeurIPS.cc/2025/Conference — NeurIPS 2025 poster_

### Official Review · Reviewer_e5xg · 2025-06-24

**Clarity:** 3
**Significance:** 2
**Originality:** 2
**Rating:** 4
**Confidence:** 4

**Summary:**

This paper presents FuncBind, an all-atom conditional generation model by representing molecular structures as neural fields, which map coordinates to continuous atomic densities. The authors evaluate FuncBind on protein target-conditioned small molecule design in 3D, antibody complementarity-determining region (CDR)  loop generation, and macrocyclic peptide (MCP) generation. FuncBind achieves competitive results on metrics when trained to generate ligands for CrossDocked protein targets, and is state-of-the-art on the SabDab dataset of protein-antibody crystal structures. This work additionally presents a new protein-MCP complex dataset containing ~190,000 synthetic complexes.

**Questions:**

1. Does one-step sampling during inference yield a speedup compared to diffusion- or flow matching-based methods? This would be an interesting point of comparison that can motivate FuncBind as a faster alternative.

2. How were the 641 complexes chosen from the PDB to build the MCP dataset?

3. Would it be possible to adapt the approach to partial molecular optimization by fixing moieties on a seed known to take part in crucial binding interactions? Fine-grained modifications to known binders is of particular interest in hit-to-lead optimization.

**Ethical Concerns:**

["NO or VERY MINOR ethics concerns only"]

**Final Justification:**

The paper presents a novel method for generating small molecules, MCP, and CDR loops in a non-equivariant manner. Despite the lack of baselines, the results, particularly in regards to inference-time speed up, are reasonably compelling.

**Limitations:**

Yes

**Quality:**

3

**Strengths And Weaknesses:**

Strengths:
1. FuncBind achieves competitive results for both small molecule design with variable functional groups and all-atom canonical and non-canonical amino acid generation, demonstrating its robustness to multiple molecular modalities.
2. FuncBind achieves state-of-the-art performance on CDR loop generation compared to other methods.
3. The model implements data augmentations rather than requiring equivariance constraints, which allows it scalability and flexibility to various architectures.


Weaknesses:
1. Although it is explained in reference [16], the authors do not define the terms "seed" or "density field" despite the concept being central to the decoding process.
2. The authors claim that the base FuncBind model leads to "high molecular validity", but 2D molecular validity metrics are not reported in any of the experiments.
3. The authors do not compare FuncBind to any baselines for the newly-introduced MCP generation task.
4. FuncBind appears to be a reasonably straightforward extension of FuncMol [16], and some of its core methodologies (e.g. sampling, loss) do not deviate extensively from those presented by FuncMol.

---

> ### Author Rebuttal · Authors · 2025-07-31
>
> We thank the reviewer for the comprehensive review and positive recognition of FuncBind's competitive performance across multiple molecular modalities and its state-of-the-art results in CDR loop generation. We also appreciate the reviewer's acknowledgment of our model's robustness and the benefit of using data augmentation over explicit equivariance constraints. Next, we address reviewers’ weaknesses and questions.
>
> **Undefined Terms ("seed," "density field").** Thank you for pointing out the missing definitions. We will clarify these terms in the main paper.
> * Seed: Refers to the original binder present in the co-crystal structure.
> * Density field: A function that maps 3D coordinates to n-dimensional atomic occupancy values, where n is the number of atom types. This continuous generalization of voxels offers advantages over point cloud representations, including compatibility with expressive neural network architectures, implicit handling of variable atom/residue counts, and an all-atom formulation across modalities.
>
> **FuncBind as an extension of FuncMol [16]:**
> While FuncBind builds upon the neural field concept introduced in FuncMol, it proposed several significant and non trivial innovations:
> * *Structure-conditioned generative model*. Unlike FuncMol, FuncBind is explicitly designed as a structure-conditioned generative model, a setting with more practical value than unconditional generation especially for drug discovery.
> * *Modality-agnosticism*. FuncMol focused primarily on small molecules and small MCPs. FuncBind is the first framework to unify all-atom generation across three distinct and challenging modalities (small molecules, MCPs, and CDR loops) within a single model, demonstrating broad applicability and leveraging diverse training data.
> * *Spatially arranged feature map*. We introduce a spatially arranged feature map (Figure 1b) in the latent space, a departure from FuncMol's global embedding. This innovation allows FuncBind to better capture local spatial information, scale to larger molecules, and integrate expressive 3D U-Net architectures for denoising.
> * *Multi-measurement walk-jump sampling (WJS-m)*. We provide the first experimental validation of WJS-m in generative modeling applications (Section 3.2.2), demonstrating its effectiveness in improving sampling by effectively decreasing noise levels upon accumulation of measurements, which is a novel contribution in this context. FuncMol was limited to single measurement walk-jump sampling which we observed has lower validity.
> * *Novel Dataset and Benchmark*. The introduction of the MCP dataset and benchmark is a substantial contribution, enabling future research in this understudied area.
>
> **2D Molecular validity metrics.** Our claim of "high molecular validity" (Lines 192-193) refers to the successful inference of chemically sensible 3D structures and bonds by OpenBabel, which inherently verifies 2D validity. We observed that increasing the number of noise levels, from walk-jump sampling to multi-measurement walk-jump sampling and ultimately to diffusion (approximating infinite noise levels), directly correlates with a higher fraction of molecules passing the OpenBabel filter. In practice, our method achieved around 50% molecular validity performance on CDRs, meaning half of our generated molecules passed the OpenBabel filter. We note that our fast sampling speed ensures that this validity rate does not become a bottleneck for the utility of our method.
>
> **Baselines for MCP.** You correctly highlight the absence of baselines for macrocyclic peptide (MCP) generation, which is precisely why we introduced a new dataset and benchmark for this challenging modality. MCPs pose significant challenges for generative models due to their non-canonical amino acids, cyclization, and scarce training data. At submission time, no other target-conditioned, structure-based MCP generative models could handle non-canonical amino acids, making direct comparisons impossible.
> To address this, we now provide a comparison of FuncBind with AfCycDesign [Rettie et al 25a] and RFPeptide [Rettie et al 25b], two very recent generative models that generate MCPs exclusively with canonical amino acids and N-to-C cyclization. As the table below shows, FuncBind achieves better or similar metrics, particularly with lower average ligand RMSDs (2.47 vs 3.08 and 3.99) and interface RMSDs (2.1 vs 3.3 and 5.2). FuncBind also yields the highest proportion of generated samples with docking scores better than the seed (50% vs 7% and 18%). The only metric where FuncBind samples underperformed was the fraction of residues with a Tanimoto similarity greater than 0.5 (0.38 vs 0.39), which was expected as FuncBind has access to a larger set of non-canonical amino acids and sequence lengths unlike the baselines in its sampling process. We encourage future comparisons of MCP models, especially those handling non-canonical amino acids, against FuncBind on this new benchmark.
> | Model         | TM-score | l-RMSD | i-RMSD | Vina Score | Tanimoto Similarity |
> |---------------|----------|--------|--------|-------------|----------------------|
> | AfCycDesign   | 0.21     | 3.08   | 3.3    | 7           | 0.39                 |
> | RFPeptide     | 0.22     | 3.99   | 5.2    | 18          | 0.30                 |
> | Funcbind      | 0.33     | 2.47   | 2.1    | 50          | 0.38                 |
>
> [Rettie et al 25a] Cyclic peptide structure prediction and design using AlphaFold2
> [Rettie et al 25b] Accurate de novo design of high-affinity protein binding macrocycles using deep learning
>
> **Sampling time**. FuncBind is not tied to any specific score-based sampling methods. In fact, we tried three generative models: single measurement and multi measurement walk jump sampling, and diffusion. All these approaches leverage several sampling steps at the different noise levels (in walk jump sampling, these sampling steps are called “walk”). Despite using several sampling steps (500 for single measurement, 1600 for multi measurement and 256 for diffusion), we still achieve the fastest sampling time, owing to our latent formulation.
> | Method                        | Target Type                | Inference Time (s/mol) |
> |-------------------------------|------------------------------|-------------------------------|
> | FuncBind-diffusion     | Small Molecule     | 2.2   |
> | FuncBind-WJS-1         | Small Molecule    | 2.1  |
> | FuncBind-WJS-16        | Small Molecule   | 5.7  |
> | VoxBind                       | Small Molecule          | 5.0                                |
> | Pocket2Mol                  | Small Molecule          | 25.4                             |
> | TargetDiff                     | Small Molecule          | 34.3                              |
> | DecompDiff                  | Small Molecule          | 61.9                             |
> | FuncBind-diffusion       | CDR                          | 2.8                               |
> | FuncBind-WJS-1           | CDR                          | 2.7                               |
> | FuncBind-WJS-16         | CDR                          | 7.3                               |
> | DiffAb                           | CDR                           | 8.9                               |
> | AbDiffuser                    | CDR                           | 34.4                             |
>
> **MCP dataset complex selection**. The 641 MCP-protein complexes are all the complexes we could isolate from the RCSB PDB [19] at the time of data curation.
>
> **Adaptability to partial molecular optimization**. Yes, our approach is adaptable to partial molecular optimization and fixing specific moieties. In fact, the antibody CDR loop generation task (Section 4.2) is a direct instance of this: it involves generating the CDR loops with a fixed antibody framework dock and epitope. Our model can be directly trained for this type of constrained generative task due to the continuous neural field representation and its ability to capture local information. As mentioned by the reviewer, this inherent flexibility makes FuncBind promising for applications like hit-to-lead optimization, where fine-grained modifications to known binders are critical.

---

> > ### Comment · Area_Chair_wYVP · 2025-08-05
> >
> > Dear Reviewer, as the deadline for this key phase of the NeurIPS review process is just a few days away, we’d greatly appreciate your engagement in any remaining discussions with the authors.

---

### Official Review · Reviewer_d5cJ · 2025-06-27

**Clarity:** 3
**Significance:** 1
**Originality:** 2
**Rating:** 5
**Confidence:** 4

**Summary:**

This paper introduces FuncBind, which improves upon VoxBind, the neural field based molecule generation method of Kirchmeyer et al., adding some improvements to the model, but also scaling to larger problems: macro-cyclic peptides and antibody CDRs.

The core contributions of this paper are:
- a demonstration of target-conditional generation
- demonstrating the robustness of a common core, applicable to different types and sizes of molecules
- a spatially-structured latent representation
- a successful application of _multimeasurement_ walk-jump sampling
- a new benchmark dataset for macrocyclic peptides

**Questions:**

Expanding on (a) and (b) I would suggest to:
- run a much more thorough comparison of VoxBind and FuncBind on the CDR and MCP tasks
- better ablate which improvements of VoxBind lead to FuncBind's increased performance; e.g. the conditioning framework, the multimeasurement (vs not) walk-jump sampling (or even just other sampling methods).
- expand on computation: I appreciate that the authors try 400M and 700M models, but it's not clear to me how that compares to other models; e.g. what is the runtime? Is the VoxBind baseline also 400M parameters? What if we scaled it? In general, the paper doesn't give a sense of what's a result of scale and more computation vs a result of better design
- expand on the dataset contribution: similarly, I appreciate that the authors intend to release a new dataset, but it feels strange to not compare FuncBind on this benchmark to any other baseline, even if somewhat trivial. Doing so would help situate that dataset in terms of how common methods perform on it, and what are the challenges still faced.

Minor: I'm not sure that FuncBind being described as "a new modality-agnostic representation" is correct, and feels a bit ambiguous. The modality is the vector field. This would feel like introducing a GNN architecture and claiming it to be a new modality-agnostic representation because everything can be a graph. Although I think I understand the intent of the authors, they may want to consider an alternative phrasing.

**Ethical Concerns:**

["NO or VERY MINOR ethics concerns only"]

**Final Justification:**

There were crucial results that were missing: some important ablations and some important comparisons to other baselines. The results provided in the rebuttal seem to support their claims and fill that gap, and in my opinion were the main missing piece from making the paper go from a "we made this work" paper to a "here's how we can advance the field" paper.

**Limitations:**

Yes

**Quality:**

3

**Strengths And Weaknesses:**

The paper's main strengths are that it demonstrates that the neural field approach is scalable, and that some clever modifications enable it to ingest more complex, spatial conditioning information. Another positive aspect of the paper is that it introduces a dataset in a domain where very little data exists.

The paper itself is well written, and I didn't have any problem understanding the method (although I was familiar with VoxBind).

The paper's main weakness is that it (a) does not appropriately compare itself to the model it is building upon, VoxBind, and that it (b) does not properly decompose the improvements that build FuncBind. This is unfortunate because it prevents us from learning and understanding which parts of the proposed method matter.

In the end, this paper reads more like a "we delivered a model"-paper than a scientific methods paper. What are the hypotheses being tested? Why or when are the proposed changes to VoxBind important? What have we learned from this paper? Other than that it's _possible_ to make performance improve.

---

> ### Author Rebuttal · Authors · 2025-07-31
>
> We thank the reviewer for the thoughtful review, particularly for recognizing FuncBind's scalability, its ability to handle complex spatial conditioning, and the significant contribution of our new macrocyclic peptide (MCP) dataset. We are also pleased that you found the paper well-written and the method clear. The reviewer's main concern is about lack of comparison with VoxBind, which we address first, followed by the other remarks.
>
> **Comparison of FuncBind and VoxBind.** Our core contribution introduces neural fields as a novel representation for structure-conditioned molecular generation, an approach previously unexplored in this context. While VoxBind is a conditional model utilizing discrete voxel grids—the discretized counterpart of neural fields—both methods share benefits like compatibility with powerful denoiser architectures and score-based generative frameworks, and ease of structure-based conditioning.
> However, the computational and memory costs of voxel representations scale cubically with molecular volume. In practice these models are only applicable to small molecule generation. This is why we did not compare FuncBind vs VoxBind on the CDR/MCP datasets: for the simple reason that VoxBind cannot be trained efficiently in these settings. Our proposed continuous neural field representation directly addresses this fundamental scalability challenge by offering significantly greater memory efficiency. This enabled its application to larger and more complex molecular systems such as MCPs and CDR loops.
>
> **This paper reads more like a "we delivered a model"-paper than a scientific methods paper. What are the hypotheses being tested? Why or when are the proposed changes to VoxBind important? What have we learned from this paper?**
> In our work, we show that continuous neural field representation combined with modern computer vision architectures and efficient score-based sampling can effectively generate diverse all-atom molecular systems conditioned on target structures, without relying on modality-specific representations or explicit equivariance constraints. Through FuncBind, we learned that modality-agnosticism is achievable with a single model spanning disparate molecular modalities (small molecules, macrocyclic peptides, antibody CDR loops), demonstrating the power of unified representation. This is highly non-trivial. Furthermore, our results further validate that explicit SE(3) equivariance constraints are not strictly necessary for high structural accuracy when strong data augmentation is employed, as evidenced by our large improvement of prior SE(3) equivariant baselines on the task of CDR loop redesign. This work also provided the first experimental validation of multi-measurement walk-jump sampling (WJS-m) in generative modeling, showing its effectiveness and efficiency for sampling.
> These learnings provide generalizable principles for potential future molecular design methodologies, moving beyond merely "delivering a model."
>
> **Ablation studies**
> we perform some additional ablation studies; first, we ablate the different sampling methods (walk jump sampling, multi measurement walk jump sampling and diffusion). We observe that adding more noise levels (from single measurement walk jump to diffusion) leads to better metrics, especially w.r.t. Strain energy.
>
> | Metric             | WJS-1 | WJS-16 | Diffusion |
> |--------------------|-------------|------------------|---------|
> | n_atoms_mean       | 13.06      | 22.9             | 20.0  |
> | vina_score_mean    | -5.09      | -6.35            | -5.78  |
> | vina_score_median  | -4.78      | -6.34            | -5.64  |
> | vina_min_mean      | -5.24      | -6.79            | -6.36  |
> | vina_min_median    | -5.05      | -6.65            | -6.16  |
> | qed                | 0.49       | 0.48             | 0.48   |
> | sa                 | 0.70       | 0.65             | 0.68   |
> | diversity          | 0.80       | 0.73             | 0.73   |
> | clashes_mean       | 4.67        | 7.4              | 7.21   |
> | se_median          | 1240        | 399              | 166     |
> Table: impact of sampling method
>
> We also include a study on the impact of the latent space’s dimension; we observe that more latent dimensions helps the model better perform loop inpainting (both higher uniqueness and metrics on unique samples).
> | Config       | Uniqueness | AAR  (unique)  | RMSD (unique)  |
> |----------------------|-------------|--------|-------|
> | code dimension 128   | 83.0   | 0.40 | 1.96  |
> | code dimension 64    | 65.5  | 0.35 | 2.11  |
>
> **Computation.** FuncBind's design, as a latent approach with continuous neural fields, inherently scales more efficiently with model size, enabling it to leverage increased computation more effectively. This contrasts with voxel-based methods like VoxBind, which face significant memory constraints that limit their scalability for larger molecules due. Thus, FuncBind's architectural choices are key to its demonstrated scalability and performance. For context, AbDiffuser reports 169M parameters, RFDiffusion ~100M, and VoxBind ~120M.
>
> **Runtime.** Below is the table of average sampling time per molecule for different methods on small molecules and antibody CDR loops with one A100 GPU. FuncBind achieves preferable runtime than other methods (sometimes by an order of magnitude). We attribute this to the efficient latent representation, to the use of standard architectures (CNN and self-attention layers only) and to the efficiency of score based generative models.
> | Method                        | Target Type                | Inference Time (s/mol) |
> |-------------------------------|------------------------------|-------------------------------|
> | FuncBind-diffusion     | Small Molecule     | 2.2   |
> | FuncBind-WJS-1         | Small Molecule    | 2.1  |
> | FuncBind-WJS-16        | Small Molecule   | 5.7  |
> | VoxBind                       | Small Molecule          | 5.0                                |
> | Pocket2Mol                  | Small Molecule          | 25.4                             |
> | TargetDiff                     | Small Molecule          | 34.3                              |
> | DecompDiff                  | Small Molecule          | 61.9                             |
> | FuncBind-diffusion       | CDR                          | 2.8                               |
> | FuncBind-WJS-1           | CDR                          | 2.7                               |
> | FuncBind-WJS-16         | CDR                          | 7.3                               |
> | DiffAb                           | CDR                           | 8.9                               |
> | AbDiffuser                    | CDR                           | 34.4                             |
>
> **MCP dataset and baselines.** You correctly highlight the absence of baselines for macrocyclic peptide (MCP) generation, which is precisely why we introduced a new dataset and benchmark for this challenging modality. MCPs pose significant challenges for generative models due to their non-canonical amino acids, cyclization, and scarce training data. At submission time, no other target-conditioned, structure-based MCP generative models could handle non-canonical amino acids, making direct comparisons impossible.
> To address this, we now provide a comparison of FuncBind with AfCycDesign [Rettie et al 25a] and RFPeptide [Rettie et al 25b], two generative models that generate MCPs exclusively with canonical amino acids and N-to-C cyclization. As the table below shows, FuncBind achieves better or similar metrics, particularly with lower average ligand RMSDs (2.47 vs 3.08 and 3.99) and interface RMSDs (2.1 vs 3.3 and 5.2). FuncBind also yields the highest proportion of generated samples with docking scores better than the seed (50% vs 7% and 18%). The only metric where FuncBind samples underperformed was the fraction of residues with a Tanimoto similarity greater than 0.5 (0.38 vs 0.39), which was expected as FuncBind has access to a larger set of non-canonical amino acids and different sequence lengths as opposed to these baselines in its sampling process. We encourage future comparisons of MCP models, especially those handling non-canonical amino acids, against FuncBind on this new benchmark.
> | Model         | TM-score | l-RMSD | i-RMSD | Vina Score | Tanimoto Similarity |
> |---------------|----------|--------|--------|-------------|----------------------|
> | AfCycDesign   | 0.21     | 3.08   | 3.3    | 7           | 0.39                 |
> | RFPeptide     | 0.22     | 3.99   | 5.2    | 18          | 0.30                 |
> | Funcbind      | 0.33     | 2.47   | 2.1    | 50          | 0.38                 |
>
> [Rettie et al 25a] Cyclic peptide structure prediction and design using AlphaFold2
> [Rettie et al 25b] Accurate de novo design of high-affinity protein binding macrocycles using deep learning
>
> **Minor: "modality-agnostic representation" phrasing**.  We will revise the current phrasing to clarify that our "modality-agnostic representation" refers to the unified approach of representing diverse molecular structures (small molecules, peptides, proteins, etc) as continuous atomic density fields, rather than using distinct, modality-specific representations (e.g., graphs for small molecules, amino sequences or backbone coordinates for proteins). This allows a single model to be trained across these varied chemical entities (and allows us to trivially model non-canonical amino acids).

---

> > ### Comment · Reviewer_d5cJ · 2025-08-01
> > **Crucial results**
> >
> > This is great, I think the new comparisons and ablations really add to the scientific strength of the paper. While this is still not an in-depth investigation of the core, in my opinion there is now sufficient evidence to make this a robust result.

---

### Official Review · Reviewer_n7Ct · 2025-06-30

**Clarity:** 2
**Significance:** 3
**Originality:** 3
**Rating:** 4
**Confidence:** 3

**Summary:**

The paper presents a single, modality-agnostic generative framework that directly produces all-atom, structure-conditioned binders in three dimensions, seamlessly spanning small molecules, macrocyclic peptides, and antibody CDR loops. It does so by representing each atom type as a continuous spatial density field and embedding these fields into a shared latent space via a variational autoencoder; generative sampling is then performed through a novel “walk-and-jump” score-based diffusion process in latent space, conditioned on the target’s binding site and the desired molecular modality. After sampling, high-resolution density grids are converted back into discrete atomic coordinates using gradient-based peak refinement and cheminformatics bond inference. Experiments on CrossDocked2020, SabDab antibody loops, and a new macrocyclic-peptide benchmark show that this unified approach achieves docking scores, amino-acid recovery rates, RMSDs, and binding-affinity improvements that are on par with or exceed those of specialized state-of-the-art methods, all without requiring separate architectures or post-hoc backbone minimization.

**Questions:**

Which components of your neural-field design are most critical for performance?

How do bond lengths, angles, and torsion profiles of your generated molecules compare to high-quality crystal structures or established force-field minima?

**Ethical Concerns:**

["NO or VERY MINOR ethics concerns only"]

**Final Justification:**

After going through the rebuttal response and other reviewer's review. I will keep suggesting accepting this paper.

**Limitations:**

Yes, with conclusion.

**Quality:**

3

**Strengths And Weaknesses:**

## Strength
The paper’s most compelling contribution is its truly unified treatment of three disparate molecular modalities--small molecules, macrocyclic peptides, and antibody CDR loops--within a single continuous-field framework. By representing each atom type as a smooth spatial density and embedding all-atom structures into a common VAE latent space, the authors avoid the need for separate graph or sequence-based architectures and specialized loss functions for each modality.

Moreover, the neural-field formalism is both elegant and flexible. Decoding via multiplicative Gabor-filtered MLPs supports arbitrary atom counts and fine spatial detail, while score-based diffusion leverages standard 3D U-Net architectures without adding the complexity of equivariant networks. This shows that high-resolution, all-atom accuracy need not rely on explicit rotation equivariance or discrete bond priors, broadening the applicability of diffusion methods to large biomolecules.

## Weakness

The decoding pipeline’s reliance on density-peak extraction, gradient ascent for atomic refinement, and OpenBabel bond inference introduces multiple heuristic stages that could each introduce errors--missing bonds, unusual torsions, or artefactual geometries. Indeed, the reported slight underperformance on strain-energy metrics versus VoxBind hints that the field representation may struggle to capture fine bond-angle or torsional constraints

The ablation studies and comparisons are insufficiently thorough. There is no systematic evaluation of how architectural choices, diffusion hyperparamters (number of Langevin steps, noise schedules), or conditioning strategies impact sample quality. Nor is there a head-to-head against recent equivariant diffusion methods on the same benchmarks. Such analyses would strengthen confidence that the unified neural-field approach is indeed the optimal design rather than a convenient simplification.

In addition to the concerns already noted, the way the empirical results are presented undermines clarity and easy comparison. For example, in the key performance table the authors have failed to highlight or bold the best numbers across methods.

---

> ### Author Rebuttal · Authors · 2025-07-31
>
> We thank the reviewer for the thoughtful review and positive assessment of FuncBind's unified, modality-agnostic approach and its innovative use of neural fields. We appreciate your recognition of our framework's ability to span diverse molecular modalities and its competitive performance against specialized baselines. Next, we address the reviewers' concerns.
>
> **Decoding pipeline's reliance on heuristic stages.** We acknowledge that our multi-stage decoding pipeline involves heuristic steps. However, the current pipeline, while not fully differentiable end-to-end for bond inference, was a pragmatic choice to enable unified, all-atom generation across diverse molecular modalities, including variable atom/residue counts and non-canonical amino acids. Our competitive results across small molecules, macrocyclic peptides, and antibody CDR loops demonstrate that FuncBind successfully generates high-quality and chemically plausible structures despite these heuristics. Finally, we also mention that many of the small molecule baselines also rely on post-processing heuristics (eg, OpenBabel to infer bonds).
>
> **Bond lengths, angles and torsions**. While it is not standard to explicitly measure bond lengths, angles, and torsions in the benchmarks we considered, these structural qualities are indirectly evaluated through metrics such as strain energy for small molecules, RMSD and IMP (related to energy) for antibodies. Notably, our diffusion-based variation improved strain energy to a level comparable with VoxBind (166 for FuncBind with diffusion vs 188 for VoxBind), and our original IMP metric remains state-of-the-art by a significant margin. This indicates that our model generates high-quality structures, performing competitively—if not better—than baselines. Furthermore, qualitative assessments reveal consistent peptide and closure bonds in macrocycles. The successful generation of novel, chemically plausible non-canonical amino acids further supports the model’s ability to learn fundamental atomic relationships.
>
> **Ablation studies**. We conducted several ablation studies during FuncBind's development. Eg, preliminary experiments led us to select the EDM2 U-Net architecture for our denoiser (over ADM), adapting its conditioning mechanism for 3D tensors.
>
> Below, we investigated various sampling configurations. First, we varied the number of sampling steps, as shown in the tables, as requested. For small molecule generation on CrossDocked, reducing sampling steps (e.g., from 256 to 64) leads to increased strain energy, while most other metrics remain stable, and clashes even improve. For antibody CDR loops, more steps generally improve performance, though uniqueness slightly decreases with increased steps.
>
> | Steps | n_atoms_mean | vina_score_mean | vina_score_median | vina_min_mean | vina_min_median | qed | sa | diversity | clashes_mean | se_median |
>  |------|--------------|------------------|--------------------|----------------|------------------|-------|-------|-----------|---------------|------------|
> | 64 | 20.15 | -5.87 | -5.72 | -6.36 | -6.17 | 0.48 | 0.67 | 0.73 | 6.87 | 300 |
> | 128 | 20.14 | -5.82 | -5.66 | -6.39 | -6.16 | 0.48 | 0.67 | 0.72 | 7.04 | 237 |
>  | 256 | 20.00 | -5.78 | -5.64 | -6.34 | -6.16 | 0.48 | 0.68 | 0.73 | 7.21  | 166 |
> Table: Impact of sampling steps on small molecule generation.
>
> | Step | uniqueness | AAR   | RMSD  |
> |------|------------|-------|--------|
> | 64   | 0.89      | 0.42 | 1.89  |
> | 128  | 0.86      | 0.44 | 1.85  |
> | 256  | 0.83      | 0.45 | 1.82  |
> Table: impact of sampling steps on CDR loop generation.
>
> Finally we perform an ablation on different sampling methods (walk jump sampling, multi measurement walk jump sampling and diffusion). We observe that adding more noise levels generally leads to better metrics, especially w.r.t. Strain energy.
> | Metric             | WJS-1 | WJS-16 | Diffusion |
> |--------------------|-------------|------------------|---------|
> | n_atoms_mean       | 13.06      | 22.9             | 20.0  |
> | vina_score_mean    | -5.09      | -6.35            | -5.78  |
> | vina_score_median  | -4.78      | -6.34            | -5.64  |
> | vina_min_mean      | -5.24      | -6.79            | -6.36  |
> | vina_min_median    | -5.05      | -6.65            | -6.16  |
> | qed                | 0.49       | 0.48             | 0.48   |
> | sa                 | 0.70       | 0.65             | 0.68   |
> | diversity          | 0.80       | 0.73             | 0.73   |
> | clashes_mean       | 4.67        | 7.4              | 7.21   |
> | se_median          | 1240        | 399              | 166     |
> Table: impact of sampling method
>
> **Components of FuncBind critical for performance**. There are two components to performance: 1) a good reconstruction performance and 2) a good denoising performance. As with most papers on latent diffusion, there is a trade-off between these two terms. Based on our experiments, the largest strength of our approach is the use of the latest EDM2 denoiser architecture which can be efficiently reused in this setting (with some adaptation from 2D to 3D data that we made to that network). Moreover, we significantly improved the reconstruction performance of our neural field based auto-encoder by using a spatial latent vector instead of a vector value latent space.
>
> **Equivariant baselines.** The vast majority of our baselines are recent equivariant diffusion models, e.g. DiffSBDD, Pocket2Mol, Molcraft, DecompDiff, TargetDiff on the small molecules and DiffAb, AbDiffuser on the antibodies.
>
> **Empirical results presentation.** We apologize for this oversight. In a revised version, we will bold the best numbers in all performance tables to enhance clarity and ease of comparison.

---

> > ### Comment · Area_Chair_wYVP · 2025-08-05
> >
> > Dear Reviewer, as the deadline for this key phase of the NeurIPS review process is just a few days away, we’d greatly appreciate your engagement in any remaining discussions with the authors.

---

> ### Comment · Reviewer_n7Ct · 2025-08-05
>
> Thanks for the response! I have no further concerns and I am willing to keep my accept score.

---

### Official Review · Reviewer_Bhid · 2025-07-01

**Clarity:** 3
**Significance:** 3
**Originality:** 2
**Rating:** 4
**Confidence:** 4

**Summary:**

This paper introduces FuncBind, a unified generative framework for structure-based drug design that generates diverse target-conditioned molecules—including small molecules, macrocyclic peptides (MCPs), and antibody CDR loops—using a single model. Its core innovation lies in a modality-agnostic neural field representation, which models molecules as continuous atomic density fields rather than modality-specific formats, enabling training across varied molecular systems while handling variable atom counts and non-canonical amino acids. FuncBind employs a conditional latent score-based model with walk-jump sampling (WJS), enhanced by a multi-measurement strategy for efficient generation. The method achieves competitive performance against specialized baselines on established benchmarks for small molecules and antibody design and introduces a new dataset of MCP-protein complexes to address the lack of data for cyclic peptides. Notably, FuncBind generates chemically plausible structures, including novel amino acids, and demonstrates that data augmentation effectively replaces explicit SE(3)-equivariance constraints, offering a scalable alternative for structure-based drug discovery.

**Questions:**

1. Please provide a critical ablation study where you compare your unified FuncBind model against modality-specific versions. How do the results of these specialist models compare to the unified model on a per-task basis? Does the unified training lead to a performance improvement, degradation, or is it neutral?

2. The introduction of a new, large-scale dataset for macrocyclic peptide generation is a fantastic contribution. How does FuncBind's performance on the MCP task compare to other methods? To establish a proper baseline, please apply at least one other relevant generative model to your new benchmark task.

3. How does FuncBind's sampling speed compare to that of other common paradigms, such as autoregressive models (e.g., Pocket2Mol) or one-shot flow-based models? A quantitative statement would help readers understand the practical trade-offs of your approach.

**Ethical Concerns:**

["NO or VERY MINOR ethics concerns only"]

**Final Justification:**

Through their rebuttal, the authors improved the manuscript's quality, demonstrated the effectiveness of their core claim via ablation studies, and refined their proposed benchmark. The main reason hindering further score improvement is that joint training on three modalities primarily enhanced generation diversity, while not significantly outperforming single-modality models in generation quality, thereby diminishing the paper's overall contribution.

**Limitations:**

yes

**Quality:**

2

**Strengths And Weaknesses:**

Strength:

1.	The paper tackles a significant and challenging problem in computational drug discovery. The goal of creating a single, unified model that can operate across different therapeutic modalities (small molecules, peptides, antibodies) is ambitious and, if successful, would represent a major step forward in the generality and applicability of generative models.

2.	The use of neural fields to represent all molecules as continuous atomic densities is an elegant approach. It naturally circumvents the challenges of varying atom/residue counts, different connectivity patterns, and diverse chemical types, providing a robust and flexible foundation for a modality-agnostic framework.

3.	The introduction of a new benchmark for Macrocyclic Peptide (MCP) generation is a contribution to the community. MCPs are a crucial therapeutic class that sits between small molecules and biologics, but they are notoriously under-resourced in terms of public datasets and benchmarks for generative modeling. This contribution will be valuable for future research in this area.

4.	Despite being a generalist model, FuncBind achieves competitive, and in some cases state-of-the-art, performance against specialized baselines (e.g., achieving excellent RMSD in CDR loop redesign). This demonstrates the viability and potential of the proposed architectural framework and unified representation.

Weakness:
The paper, despite its strengths, suffers from several critical weaknesses that undermine its core claims and the solidity of its contributions.

1. The central hypothesis is unsubstantiated. The paper's core premise is that a unified model is superior because it can "learn physical properties across diverse atomic systems." This claim of synergistic learning or knowledge transfer is the primary justification for the entire unified approach. However, this hypothesis is not tested or validated. The paper fails to address fundamental questions:

    * What are the specific challenges of training on such diverse modalities simultaneously (e.g., conflicting optimization signals, modality imbalance, different length/size scales)? How does the model design address them?

   * Is there any evidence of positive knowledge transfer? For instance, does training on small molecules and their intricate atomic interactions improve the model's ability to generate physically plausible CDR loops?
Without this analysis, the "unification" feels more like a simple concatenation of datasets rather than a principled approach. The motivation remains unproven.

2. Lack of essential ablation studies. To validate the central hypothesis, the most crucial and necessary experiment is a direct comparison between the unified model and single-modality versions of the same model. The authors should have compared the performance of:
    * FuncBind (Unified): Trained on all three modalities, as presented.
    * FuncBind (Small Molecule only): Trained and evaluated only on CrossDocked2020.
    * FuncBind (CDR only): Trained and evaluated only on SAbDab.
    * FuncBind (MCP only): Trained and evaluated only on the new MCP dataset.

This ablation is fundamental. The current results only show that the FuncBind architecture is capable of handling these tasks, not that the unified training strategy itself provides any advantage. This is a major flaw in the experimental design.

3. Marginal or inconsistent performance gains over specialized baselines. Across Tables 1 and 2, FuncBind's performance gains over existing specialist models are not consistently clear, which weakens the argument for its superiority.
    * In Table 1 (Small Molecules): On this highly competitive benchmark, FuncBind achieves performance that is on par with, but not superior to, leading models like VoxBind and MolCraft. For instance, its Vina docking scores are slightly worse, and its Clash and Strain Energy metrics are less favorable. By only matching the performance of its direct predecessor (VoxBind), it raises the question: what is the benefit of the added complexity and unification if it does not lead to demonstrably better small molecule generation?
    * In Table 2 (CDR Redesign): The story is more nuanced but still not an unqualified win. FuncBind shows a remarkable and significant advantage in structural fidelity (RMSD), which is a major achievement. However, its performance on other key metrics like Amino Acid Recovery (AAR) is inconsistent, sometimes outperforming but often lagging behind the baseline AbDiffuser. The authors should try to explain this.

4. Insufficient Evaluation of the New MCP Benchmark: While the creation of the MCP benchmark is a commendable effort, its evaluation is critically incomplete. A "benchmark" by definition requires comparative data to be useful. In Table 3, the authors only report the performance of their own model, FuncBind, without a single baseline. This makes the reported metrics (e.g., TS=0.29, L-RMSD=2.19, Vina Dock=42%) completely uninterpretable. Are these numbers good or bad? The authors should have adapted at least one or two relevant protein generative models (e.g., RFDiffusion, which they cite as a key related work for peptide design) to their new task to provide a reference point. As it stands, the value of this contribution is diminished as it fails to serve its purpose as a benchmark.

5. The technical components of FuncBind are largely an integration of existing methods: neural fields for molecular representation are from [16], score-based generation and WJS are from [17, 64]. While applying this combination to the novel problem of multi-modality generation is a contribution, its significance is tied to the justification for doing so. As highlighted in Weaknesses 1 and 2, this justification is missing. The contribution thus appears to be more of an engineering integration rather than a fundamental methodological advance that has been proven necessary or superior.

6. The paper completely omits any discussion of sampling efficiency. Score-based models, especially those requiring many iterative sampling steps like WJS-m, are known to be computationally expensive. For practical drug discovery, where high-throughput virtual screening is paramount, generation speed is a critical metric of utility. A report on the wall-clock time to generate a batch of molecules, or a comparison with faster methods (e.g., one-step flows or autoregressive models), is essential for a comprehensive evaluation of the model's practicality. This omission leaves a gap in understanding the model's real-world viability.

---

> ### Author Rebuttal · Authors · 2025-07-31
>
> **Unsubstantiated central hypothesis**. We appreciate the feedback regarding our central hypothesis and the call for an ablation study comparing unified and modality-specific models. This analysis is insightful and we have now included an ablation study to directly address this point.
>
> Unified training is inherently challenging, which explains the prior absence of models spanning the three modalities we consider. Our neural-field-based representation, however, made multi-modality training seamless and straightforward; we achieved this by simply conditioning the denoiser on the modality class and applying uniform rebalancing. While performance on individual saturated benchmarks could be further optimized with modality-specific tuning, our primary goal was to establish a robust unified model, not necessarily to achieve state-of-the-art on metrics known to poorly correlate with true binding.
>
> The appeal of multi-modality lies in its potential for positive knowledge transfer, enabling better generalization by leveraging more diverse data and eliminating the need for separate models. Given the distinct nature of our modalities, evaluating this transfer is particularly insightful. To empirically address the reviewer's concern, we trained modality-specific models for small molecule, CDR loop, and MCP generation, and the results are compared against our unified model in the revised manuscript.
>
> **Ablation: unified vs. single-modality**. We conducted this ablation using similar compute resources and diffusion as the score-based generation method. The comparison considers uniqueness/diversity and performance metrics across modalities:
>
> * *CDR loop inpainting*: The specialized model shows a drop in sequence design uniqueness, with comparable performance on unique samples. Given uniqueness's importance in library design, the unified model's ability to maintain higher uniqueness is a notable advantage.
>
> | Loop | Uniqueness Unified | Uniqueness Specialized |
> |--|--:|--:|
> |H1|10.1|9.6|
> |H2|15.6|12.6|
> |H3|83.0|69.2|
> |L1|19.8|10.6|
> |L2|11.2|14.0|
> |L3|34.7|22.3|
>
> For the H3 loop specifically on unique samples:
> | H3 loop | AAR | RMSD |
> |--|--|--|
> | Unified | 0.399 | 1.96 |
> | Specialized | 0.406 | 2.07 |
>
> * *Small Molecules (CrossDocked)*: Metrics remained largely comparable between unified and specialized models, indicating no significant performance degradation from unification.
>
> | Configuration | n_atoms_mean | vina_score_mean | vina_score_median | vina_min_mean | vina_min_median | qed | sa | diversity | clashes_mean | se_median |
> |--|--|--|--|--|--|--|--|--|--|--|
> | Unified | 20.00  | -5.78 | -5.64  | -6.34 | -6.16  | 0.48 | 0.68 | 0.73  | 7.21 | 166 |
> | Specialized | 19.68 | -5.70  | -5.64 | -6.29  | -6.09  | 0.50 | 0.67 | 0.72 | 7.20 | 194 |
>
> * *Macrocyclic Peptides (MCP)*: The unified model demonstrated superior performance in key structural metrics (TM-score, lRMSD, iRMSD) and cyclization fraction, with other metrics remaining similar.
>
> | Configuration | lRMSD | iRMSD | TM_score | TS_full | Ratio_>0.5_TSPR | cyclization_frac |
> |--|--|--|--|--|--|--|
> | Unified | 2.65 | 1.85 | 0.33 | 0.35 | 0.27 | 0.75 |
> | Specialized | 2.90 | 1.98  | 0.30 | 0.36 | 0.27 | 0.73  |
>
> In summary, the unified model generally shows a positive to neutral effect on performance across modalities. A key benefit is the improved uniqueness, likely due to mitigating overfitting from a larger, more diverse training set. Furthermore, unification greatly simplifies experimental validation by eliminating the need to train separate models per modality. We note that transfer is likely to have a greater impact between more similar molecular classes (e.g., antibodies and proteins); this is an area for future work.
>
> **Performance on benchmarks.**
> * *Small molecules*: We acknowledge that FuncBind's performance on the highly competitive CrossDocked benchmark is on par with, or slightly behind, leading specialized models like VoxBind. However, achieving this parity while being a unified model capable of tackling significantly larger and more complex modalities like macrocyclic peptides and CDR loops—which VoxBind's discrete voxel representation inherently struggles with due to cubic scaling limitations —is a major achievement for generalizability and viability. Our primary goal was to demonstrate unification without significant performance compromise on individual tasks, rather than solely achieving state-of-the-art on (saturated) benchmarks.
> * *CDR redesign*: We appreciate your recognition of FuncBind's "remarkable and significant advantage in structural fidelity (RMSD)". This is indeed an unqualified win. Our Amino Acid Recovery (AAR) is competitive, and with recent enhancements using a diffusion model, FuncBind now leads on all AAR metrics as shown in the updated table below. Furthermore, our strong IMP (interface energy improvement) scores in the original manuscript, nearly double that of some baselines without explicit minimization, underscore the quality of our generated structures.
>
> |Loop|FuncBind AAR|FuncBind RMSD|AbDiffuser AAR|AbDiffuser RMSD|DiffAb AAR|DiffAb RMSD|
> |----|-----------|------------|--------------|----------------|---------|------------|
> |H1|88.0|0.36|76.3|1.58|65.8|1.19|
> |H2|79.7|0.44|65.7|1.45|49.3|1.08|
> |H3|44.8|1.82|34.1|3.35|26.8|3.60|
> |L1|91.0|0.58|81.4|1.46|55.7|1.39|
> |L2|91.2|0.82|83.2|1.40|59.3|1.37|
> |L3|82.9|0.60|73.2|1.59|46.5|1.63|
> *Table*: Comparison of Funcbind with diffusion model against baselines.
>
> * For practical validation beyond in silico metrics, we tested our antibody H3 redesign in the wet-lab using Surface Plasmon Resonance (SPR) in a de novo setting. This yielded a 42% binding rate on a target with a rigid epitope and 4% on a target with a flexible epitope. Further details will be provided in future revisions, pending legal clearance.
>
> **MCP baselines**. You correctly highlight the absence of baselines for macrocyclic peptide (MCP) generation, which is precisely why we introduced a new dataset for this modality. MCPs pose significant challenges for generative models due to their non-canonical amino acids, cyclization, and scarce training data. At submission, no other target-conditioned, structure-based MCP generative models handled non-canonical amino acids, precluding direct comparisons.
> To address this, we now compare FuncBind with AfCycDesign [Rettie et al 25a] and RFPeptide [Rettie et al 25b], two models generating MCPs exclusively with canonical amino acids and N-to-C cyclization. As shown below, FuncBind achieves superior or similar metrics, notably lower average ligand RMSDs (2.47 vs 3.08 and 3.99) and interface RMSDs (2.1 vs 3.3 and 5.2). FuncBind also yields the highest proportion of designs with better docking scores (50% vs 7% and 18%). The only underperformance was in Tanimoto similarity (0.35 vs 0.39), expected as FuncBind accesses a larger set of non-canonical amino acids and different sequence lengths as opposed to these baselines. We encourage future comparisons on this new benchmark, especially for models handling non-canonical amino acids.
> |Model|TM-score|l-RMSD|i-RMSD|VinaScore|TanimotoSimilarity|
> |-----|--------|------|------|----------|------------------|
> |AfCycDesign|0.21|3.08|3.3|7|0.39|
> |RFPeptide|0.22|3.99|5.2|18|0.30|
> |Funcbind|0.33|2.65|1.85|50|0.35|
>
> [Rettie et al 25a] Cyclic peptide structure prediction and design using AlphaFold2
> [Rettie et al 25b] Accurate de novo design of high-affinity protein binding macrocycles using deep learning
>
> **Technical components**. We respectfully disagree that FuncBind is merely an integration of existing methods. Our core contribution is a novel neural field representation specifically for structure-conditioned generation, a previously unexplored area. This enabled us to develop one of the first unified models for generating small molecules, MCPs (with non-canonical AAs), and CDR loops, all conditioned on a target structure, achieving competitive or SoTA results across diverse chemical matter. The unified training, facilitated by this representation, was a direct byproduct of its inherent flexibility.
>
> Unlike the unconditional FuncMol, FuncBind is explicitly structure-conditioned. It introduces significant methodological advancements like replacing FuncMol's vector latent with a spatial latent tensor, which greatly improves performance by enabling U-Net architectures and simplifying structure conditioning. Finally, while FuncBind utilizes established score-based generative models (walk-jump sampling in the initial draft, and diffusion in revision), this is common practice in the field. Our primary aim was not to invent a new sampler, but to propose a new, highly compatible representation for structure-conditioned generation that can seamlessly integrate with any score-based generative framework.
>
> **Sampling efficiency**. We agree that sampling efficiency is a crucial practical metric. FuncBind demonstrates a significant speed advantage, sometimes an order of magnitude faster than many baselines, as shown in the table below (average time in seconds per molecule on A100).
>
> |Method|TargetType|InferenceTime(s/mol)|
> |------|-----------|--------------------|
> |FuncBind-diffusion|SmallMolecule|2.2|
> |FuncBind-WJS-16|SmallMolecule|5.7|
> |VoxBind|SmallMolecule|5.0|
> |Pocket2Mol|SmallMolecule|25.4|
> |TargetDiff|SmallMolecule|34.3|
> |DecompDiff|SmallMolecule|61.9|
> |FuncBind-diffusion|CDR|2.8|
> |FuncBind-WJS-16|CDR|7.3|
> |DiffAb|CDR|8.9|
> |AbDiffuser|CDR|34.4|
>
> Unlike FuncBind, these slower models may face more issues with real-world viability where high-throughput generation is critical. While lower sampling time is useful, we prioritize models with demonstrated high experimental binding rates, a benchmark FuncBind has achieved in CDR loop redesign.

---

> > ### Comment · Reviewer_Bhid · 2025-08-04
> >
> > The author addressed most of the concerns through detailed responses, and I will increase the score accordingly.

---

### Decision · Program_Chairs · 2025-09-17

**Decision:**

Accept (poster)

**Comment:**

Summary and Strengths: 1. This paper presents an ambitious and significant advance by creating a unified model that generates diverse therapeutic molecules. 2. Its elegant approach uses neural fields to represent molecules continuously, overcoming key structural challenges.  3. The model achieves competitive, sometimes state-of-the-art, results against specialized alternatives and provides a valuable new benchmark for the under-resourced area of macrocyclic peptides.

Weaknesses: 1. The paper's central hypothesis lacks support. 2. Critical ablation studies are absent. 3. The performance gains over specialized baselines are marginal or inconsistent. 4. The clarity of the empirical results is undermined by poor presentation and a lack of clear definitions for central concepts.

The paper is recommended for acceptance as the authors have satisfactorily addressed all major reviewer concerns, leading to a consistent consensus for acceptance based on the paper's valuable contributions.